# Polymeric Micro/Nanocarriers and Motors for Cargo Transport and Phototriggered Delivery

**DOI:** 10.3390/polym13223920

**Published:** 2021-11-12

**Authors:** Pedro Mena-Giraldo, Jahir Orozco

**Affiliations:** Max Planck Tandem Group in Nanobioengineering, Institute of Chemistry, Faculty of Natural and Exact Sciences, University of Antioquia, Complejo Ruta N, Calle 67 # 52-20, Medellin 050010, Colombia; pedro.mena@udea.edu.co

**Keywords:** photosensitive micro/nanocarriers, polymeric nanomicelles, polymeric nanocarriers, nanopolymersomes, cargo photodelivery, targeting cells, light-stimulated polymeric micro/nanomotors

## Abstract

Smart polymer-based micro/nanoassemblies have emerged as a promising alternative for transporting and delivering a myriad of cargo. Cargo encapsulation into (or linked to) polymeric micro/nanocarrier (PC) strategies may help to conserve cargo activity and functionality when interacting with its surroundings in its journey to the target. PCs for cargo phototriggering allow for excellent spatiotemporal control via irradiation as an external stimulus, thus regulating the delivery kinetics of cargo and potentially increasing its therapeutic effect. Micromotors based on PCs offer an accelerated cargo–medium interaction for biomedical, environmental, and many other applications. This review collects the recent achievements in PC development based on nanomicelles, nanospheres, and nanopolymersomes, among others, with enhanced properties to increase cargo protection and cargo release efficiency triggered by ultraviolet (UV) and near-infrared (NIR) irradiation, including light-stimulated polymeric micromotors for propulsion, cargo transport, biosensing, and photo-thermal therapy. We emphasize the challenges of positioning PCs as drug delivery systems, as well as the outstanding opportunities of light-stimulated polymeric micromotors for practical applications.

## 1. Introduction

Micro/nanotechnology manipulates matter at the micro/nanoscale, resulting in enhanced properties concerning the bulk material counterparts, which are exploited in multipurpose assemblies, with significant impact in various fields [1,2,3]. For example, engineering, physics, chemistry, biology, pharmacy, and biomedical sciences converge in the development of static (S) and dynamic (D) PCs, which have stimulated the scientific community’s interest in recent years [4,5,6,7,8,9,10]. SD-PCs can be designed to exhibit biocompatibility [11,12], reproducibility [13,14], and malleability [15,16] in various morphologies [17,18], providing great versatility [19,20]. They can have a high cargo-loading capacity and transport and delivery capabilities while protecting the activity of multiple cargoes, such as plant extracts, drugs, proteins, DNA (RNA), and photosensitizers [21,22,23,24,25]. Moreover, they can interact with various media, cellular and biological environments, pollutants, and light stimuli, impacting the efficiency of environmental and biomedical interventions [26,27,28,29]. 

The first cargo release systems and propulsion mechanisms explored consisted of cargo diffusion across the carriers and fuel–motor interactions, requiring longer delivery times, but producing a synergic effect among chemical reactions for dynamic motion [30,31]. The exploration of new concepts is leading to the acceleration of the cargo release kinetics and the achievement of fuel-free movement [32]. Of particular interest is the attainment of spatiotemporal control over SD-PCs, in order to stimulate and enhance their therapeutic effects, as well as their (bio)sensing and remediation duties, with light offering outstanding opportunities for this purpose [33,34,35]. Stimuli-responsive polymers react to changes in pH, temperature, salt concentration, light, and solvent medium, among others, accelerating the cargo delivery systems, along with fast-response applications in textiles, sensors, and tissue engineering, such as thermoresponsive polymer (TRP)–additives and TRP–protein interactions, for drug delivery and technological/industrial approaches, respectively [36,37].

Energy sources, lasers, lamps, and chemical and electrochemical reactions can generate light [38,39,40,41]. Light is electromagnetic radiation, made up of photons, characterized to be in the spectral region between the UV and NIR wavelengths (200–1000 nm), and exhibits the property to transit in the void, some media, liquids, and even human tissues, among others [42,43]. Control of the power, beam diameter, wavelength, exposition area, frequency, and exposure time of light enables a diversification of purposes, such as photodynamic therapy, skin treatment, surgical procedures, imaging techniques, disinfection, and pollutant remediation, among others [44,45,46,47,48,49]. The field of optical physics studies the control of light properties, along with multiple techniques to produce a specific wavelength [50,51]. For example, a light-emitting diode (LED) and a tungsten lamp source can produce UV and NIR irradiation, respectively [52,53]. UV light has low tissue penetration but high photostimulation, phototoxicity, disinfection, and chemical decomposition induction capabilities, in contrast to NIR irradiation [54,55].

Taking advantage of the properties, versatility, and control of light, photoactive molecules—such as azobenzenes, photosensitizers and ortho-nitrobenzene; gold, upconversion nanoparticles (UCNPs), among others—have been used to produce photosensitive polymeric micro/nanocarriers and motors with the versatility of structures to respond to light stimuli [56,57,58,59]. They provide high external and non-invasive spatiotemporal control, accelerating the cargo release, activating photoreactions, fuel-free, and self-motion of micro/nanomotors via photo-isomerization, photo-cleavage, and thermophoresis mechanisms, which are of importance in the field of biophotonics [60,61,62,63].

This review shows an overview of the progress in photosensitive polymer synthesis and micro/nanocarrier assembly, cargo transport, and photodelivery in biomedical applications, including their functionalization with cell-targeting biomolecules for specific cellular uptake (Figure 1A). It also highlights current advances in light-stimulated polymeric micro/nanomotors, stressing their potential for practical applications (Figure 1B).

## 2. Organic and Polymeric Micro/Nanocarriers

Micro/nanocarriers are organic and polymeric materials that are structurally oriented to act like capsules in aqueous and organic media in order to protect, transport, and release cargo [64,65,66,67], among other applications. Liposomes are organic micro/nanocarriers based on phospholipids that form vesicles [68]. In contrast, polymeric micro/nanocarriers are based on amphiphilic and backbone polymers that form micelles [69], polymersomes [70], and polymeric spheres [71]. Micro/nanocarriers are usually produced via precipitation and emulsion techniques, using a hydrophobic/hydrophilic solvent mixture, an emulsifier or surfactant, and crosslinker agents acting as template-like reaction initiators to orientate the amphiphilic backbone polymers, forming their structure [72,73,74]. Cargo is loaded either during the assembly process, by solubilizing the cargo in the hydrophobic or hydrophilic solvent, or after the carrier’s formation, by dispersing the carriers into a high-cargo-concentration solution, followed by further cargo diffusion through the carrier [75,76].

Cargo is released by diffusion and osmotic pumping mechanisms, as well as polymeric matrix erosion and degradation [77]. Unbalanced external/internal cargo concentration produces a concentration gradient with the loaded cargo and, consequently, its release via diffusion or osmotic pressure mechanisms through the water-filled cavities, polymeric matrix, and nanostructural porosity of the micro/nanocarriers [78,79,80]. Conversely, hydrolysis in the medium and external stimulus causes polymeric carrier erosion and degradation, delivering the cargo [81]. Furthermore, the cargo release time is directly affected by π-π electrostatic and adsorption interactions between the cargo and the polymeric matrix, preventing immediate delivery [82]. In the same manner, cargo precipitation in the medium caused by changes in the temperature, pH, and polarity can act as barriers in the release mechanism, causing slower delivery kinetics [83]. Superficial fluid velocity improves the cargo delivery, because it is removed from the surface faster via convection, preventing saturation and increasing the concentration gradient for a constant cargo release [84]. The polymeric micro/nanocarriers’ features and properties—such as their permeability, thickness, and polymer and carrier sizes—play an essential role in the cargo loading/release time, because these characteristics and the cargo release kinetics are inversely related [85].

The transport, protection, and delivery of the cargo are strategies for drug delivery [86], enzymatic activity protection [87], bioremediation [88], biosensing [89], and medical intervention [90]. Reproducibility, flexibility [91], malleability [92], and biocompatibility [93] of those based on polymeric micro/nanocarriers make them promising candidates for developing the next generation of micro/nanocarriers [94]. Principal characteristics and representative applications of micro/nanocarriers are shown as follows.

### 2.1. Liposomes

Liposomes are synthetic micro/nanovesicles composed of phospholipid bilayers with a hydrophilic core, a hydrophobic bilayer, and a hydrophilic shell [95] (Figure 2(Aa)). The aqueous internal core of the liposome can encapsulate polar cargo, while the lipid bilayer can incorporate apolar cargo, providing many possibilities for amphiphilic cargo encapsulation [96]. Nanoliposomes can release the cargo for intracellular drug delivery, because they may directly interact with the external cellular membrane, followed by cellular uptake [97]. The cargo is released intracellularly, either passively via liposome disruption, or via disintegration after fusion with cell membranes [98]. For example, Guo et al. developed a nanoliposome encapsulating ferric ammonium citrate in the hydrophilic core, transporting and releasing the cargo to a rat brain, thus increasing the local iron concentration for iron-deficiency anemia therapy (Figure 2A) [99].

### 2.2. Polymeric Micro/Nanomicelles

Copolymer micro/nanomicelles (MNMs) are composed of two polymer blocks with different hydrophilic/hydrophobic natures [103]. In aqueous environments, these MNMs spontaneously assemble into a core–shell-like structure (Figure 1(Ab)). The hydrophobic segment aggregates integrate in the core, while the hydrophilic segment segregates the shell block [104], showing a high distribution and high hydrophobic cargo-loading capacity into the core [105]. For example, in a strategy recently developed by our group, polymeric nanomicelles based on an amphiphilic poly (lactic acid-co-glycolic acid) (PLGA) polymer were self-assembled. The nanomicelles were formed via the nanoemulsion method, with a hydrophobic core and a hydrophilic shell (Figure 2B), charging itraconazole with high loading capacity and encapsulation efficiency, taking advantage of the drug–core hydrophobic nature. The nanomicelles’ surface was further functionalized with an F4/80 antibody and mannose via the adsorption and carbodiimide methods, for an enhanced uptake by macrophages. The itraconazole released specifically into the cells from the nanomicelles, following a Fickian diffusion mechanism, and demonstrating enhanced efficacy and efficiency in eliminating the *Histoplasma capsulatum* fungus as a drug delivery strategy to fight intracellular infections [100]. On the other hand, different micelle properties have been investigated to improve the cargo release effect, such as critical micellization temperature (CMT) and the micelle formation’s minimum temperature. For example, Umapathi et al. studied a smart-responsive triblock copolymer based on poly(ethylene glycol) (PEG)-poly(propylene glycol)-PEG to understand the behavior of the CTM, with specific anion effects, which are directly related to the drug delivery mechanisms [106].

### 2.3. Micro/Nanopolymersomes

Micro/nanopolymersomes are amphiphilic-membrane bilayer vesicles with a hydrophilic core, hydrophobic interphase, and a hydrophilic shell (Figure 1(Ab)), permitting oily and aqueous cargo encapsulation and transport in aqueous solutions [107]. Co-solubilization in hydrophobic and hydrophilic solvents promotes structure formation via self-assembly, followed by elimination of hydrophobic solvents via evaporation and dialysis [108]. Polymersomes provide a dual hydrophobic/hydrophilic cargo transport alternative due to having a hydrophilic core and a hydrophobic interface [109]. Furthermore, the cargo release kinetics are controlled by varying the thickness of the hydrophobic interface, where it being thinner promotes a faster cargo release, but its hydrophobic loading capacity is lower [110]. As a way of illustrating such a dual carrying capability, Khan et al. synthesized nanopolymersomes for anticancer drug release, based on a poly((mPEG-SS-amino) (N,N-diisopropylethylenediamino)phosphazenes). Nanopolymersomes loaded hydrophobic and hydrophilic anticancer drugs in the hydrophilic core and hydrophobic interface, respectively (Figure 2C), providing a nanoplatform for dual drug delivery [101].

### 2.4. Polymeric Micro/Nanospheres

Unlike micro/nanomicelles and micro/nanopolymersomes, polymeric micro/nanospheres are structures characterized as having the backbone polymer agglomerated to form solid carriers with the ability to transport cargo with different polarities (Figure 1(Ab)) [111]. Nanoprecipitation and crosslinking methodologies are used to form polymeric micro/nanospheres, producing a solid polymeric micro/nanocarrier via polymer self-dispersion or crosslinking of available terminal groups by chemical reactions. Solid spheres have a long cargo release timeframe and cargo immobilization ability, enabling a compact structure with long life to prevent rapid degradation [112,113]. Ceron et al. synthesized polymeric microspheres based on a chitosan polymer crosslinked with glutaraldehyde (Figure 2D). Microspheres immobilized lysozyme via the aldehyde available in the polymer and the free amines from the enzyme. The lysozyme-linked microspheres efficiently produced the lysis of *Micrococcus lysodeikticus*, demonstrating antimicrobial activity by inhibiting *Staphylococcus aureus*, *Enterococcus faecalis*, and *Pseudomonas aeruginosa* bacteria [102].

## 3. Photosensitive Polymeric Micro/Nanocarriers

Photosensitive polymeric micro/nanocarriers (PMNs) are carrier systems that respond to light stimuli, altering their physicochemical structure, leaving cargo release with a spatiotemporal control [114]. According to the carrier configuration, the PMNs are classified as micelles, polymersomes, and spheres [115]. A strategy to synthesize PMNs is to introduce photoactive molecules (PAMs) into a polymer backbone to produce photosensitive amphiphilic polymers, and to use this characteristic to form micro/nanocarriers [116]. UV, visible, and NIR irradiation photostimulate PAMs, causing photodimerization, photocyclization, photogeneration of reactive species, photolysis, photocleavage, and photoisomerization. Such stimuli may change their physicochemical characteristics, destabilizing the PMNs’ amphiphilicity and releasing the cargo with accelerated kinetics and specific distribution [117,118]. This review focuses on photocleavage and photoisomerization, because they are the more characteristic photostimulation mechanisms. Optical, fluorescence, and transmission electron microscopy (TEM), scanning electron microscopy (SEM), Fourier-transform infrared spectroscopy (FT-IR), nuclear magnetic resonance (NMR), dynamic light scattering (DLS), static light scattering (SLS), electrophoretic light scattering (ELS), and gravimetric techniques, among others, characterize nanocarriers’ morphology, physicochemical properties, loading capacity (LC), encapsulation efficiency (EE), and cargo photorelease [119,120]. PMN-based strategies seek to control the cargo release for diagnosis [121], tissue engineering [122], drug delivery [123], targeted pesticides [124], etc. Cargo release with accelerated kinetics is a typical application of PMNs to improve the therapeutic effect in treating diseases [125], and spatiotemporal control of PMN-based technologies offers an attractive alternative for cargo delivery [34].

### 3.1. Photoisomerizable and Photocleavable Molecues

UV light isomerizes UV-PAMs such as azobenzenes and spirobenzopyrans from trans to cis, changing their resonant structure [126], while visible light reverts the UV-PAMs cis isomer to a trans one [127]. Azobenzene and spirobenzopyran molecules’ structure and photoisomerization are represented in Table 1A,B, respectively. The azobenzene is photoexcited to the S_1_ (n → π*) state around the N=N double bond, producing their cis and oxidated form (Table 1A) [128]. The spirobenzopyrans suffer a photoreaction opening their specific ring via heterolytic C-O bond cleavage, changing the form of the isomer (Table 1B) [129].

The UV light and the NIR irradiation excite UV–NIR PAMs such as ortho-nitrobenzenes, pyrenylmethyl ester, and coumarinyl ester, generating anti-Stokes emissions and breaking specific covalent bonds [130,131,132]. The UV–NIR PAMs’ molecular structures, photocleavage mechanisms, and oxidated forms are depicted in Table 1C–E, respectively. The ortho-nitrobenzene is photo-oxidated, and the nitro group is reduced to nitroso, causing covalent bond rupture (Table 1C) [133]. The pyrenylmethyl ester suffers the cleavage in the ester group as a result of photoinduced hydrogenation (Table 1D) [134]. The coumarinyl ester is photoexcited, forming the hydroxymethyl coumarin and delivering the protected carboxylic acid (Table 1E) [135].

**Table 1 polymers-13-03920-t001:** Groups, structures, and oxidation processes of the PAMs [127,136].

	Groups	Structures	Oxidation Processes
**A**	Azobenzene	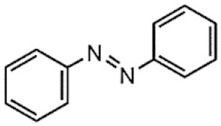	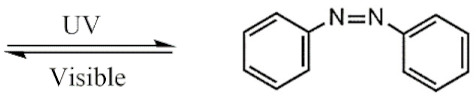
**B**	Spiropyran	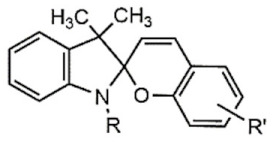	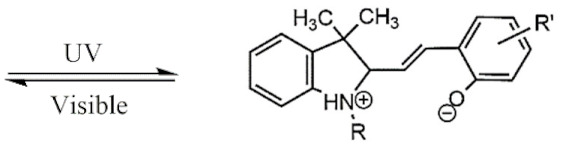
**C**	O-nitrobenzyl ester	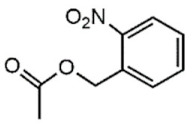	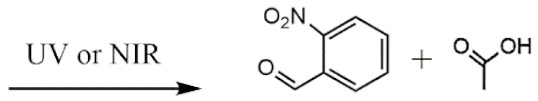
**D**	Pyrenylmethyl ester	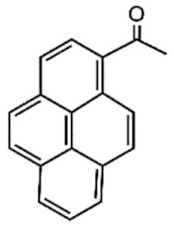	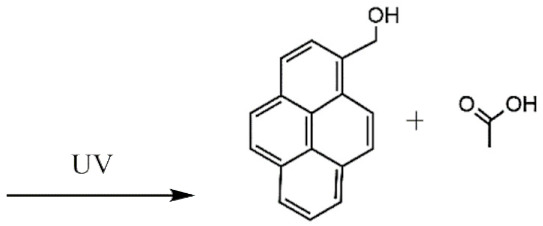
**E**	Coumarinyl ester	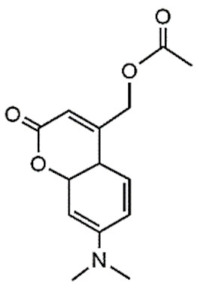	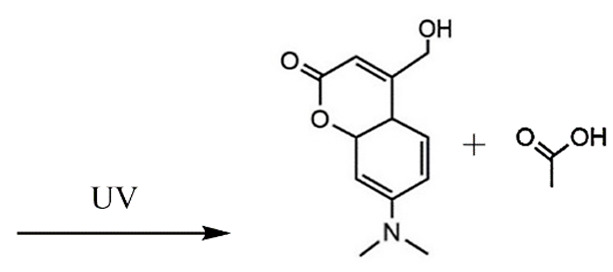

### 3.2. Photoactive Polymers: Synthesis and Assembly

PAMs are linked to polymeric backbones via addition and substitution reactions, conditioning the PAMs’ terminal groups for their availability to produce UV photoisomerizable and UV–NIR photocleavable amphiphilic polymers [137]. The PAMs have hydrophobic characteristics, i.e., the apolar segment of the produced amphiphilic polymer or the hydrophobic/hydrophilic bridge-like segments (hydrophobic-b-hydrophilic) (Figure 1(Aa)) [138]. This polymer amphiphilicity may lead to various PMNs—such as nanomicelles [139] (Figure 2B), nanopolymersomes [140] (Figure 2C), and nanospheres [141]—using nanoprecipitation [142], nanoemulsion [143], crosslinking [144], and layer- by-layer [145] techniques for their assembly (Figure 1(Ab)). For example, Perez-Buitrago et al. synthesized a photoisomerizable poly2-[4-phenylazophenoxy]ethyl acrylate-co-acrylic acid (PPAPE) amphiphilic copolymer, functionalizing an azobenzene molecule (Table 1A), modified with a terminal hydroxyl group with an acryloyl chloride polymer via nucleophilic substitution. After hydrolysis, the azobenzene hydrophobic segments and the carboxyl hydrophilic segments formed an amphiphilic polymer—and, consequently, nanomicelles—utilizing the nanoprecipitation method with tetrahydrofuran (THF) and water solvents [142]. Similarly, Figure 3 shows some schematic synthetic route representations of UV photoisomerizable polymers and their nanocarrier structures. For example, Mena-Giraldo et al. developed a UV-photosensitive N-succinyl-chitosan (PNSC) via an amidation reaction between carboxylic groups of modified azobenzene and the free amines of the N-succinyl-chitosan, as previously synthesized. This produced an amphiphilic polymer with the azobenzene group as an apolar section and carboxylic groups as polar ones, employing the nanoprecipitation method with THF/H_2_O to form the nanomicelles (Figure 3A) [146].

In the same way, Molla et al. designed a novel assembly technique wherein the azobenzene group works as a linker between PEG and polylactic acid (PLA), via the esterification of hydroxyl groups as the principal reaction. This resulted in an amphiphilic PEG-b-PLA polymer block, which formed nanopolymersomes via nanoprecipitation (Figure 3B) [147]. On the other hand, Razavi et al. utilized a backbone and amphiphilic polymer with terminal hydroxyl (OH) groups—i.e., poly (N-isopropylacrylamide)-b-poly (methyl methacrylate) (PNIPAM-b-PMMA)—to be linked in a side with a spiropyran (SP) (Table 1B). The SP was modified with a terminal bromide and linked to the polymer for a nucleophilic substitution of Br for OH. Similarly, the modified SP was linked to the other side of the polymer, generating two amphiphilic polymers—i.e., SP-(PMMA-b-PNIPAM) and SP-(PNIPAM-PMMA)—which can form nanomicelles via the nanoprecipitation method (Figure 3C) [148].

Figure 4 represents some usual synthetic routes of UV–NIR photocleavable polymers and their carrier assemblies. For example, Cao et al. synthesized an amphiphilic photosensitive N-succinyl-chitosan (APNSC) polymer via amidation between the photo-cleavable o-nitrobenzyl group (Table 1C) and the N-succinyl-chitosan backbone. The o-nitrobenzyl was modified with a carboxyl terminal group and activated via carbodiimide methodology to link with the available amines from the backbone polymer. These PAMs were the hydrophobic segment of the polymer, while the carboxyl groups were hydrophilic, enabling the authors to obtain nanomicelles via the nanoemulsion technique (Figure 4A) [149]. In the same way, Doi et al. functionalized 4,5-dimethoxy-2-nitrobenzyl methacrylate (DNMA) as a hydrophobic segment in a particular poly[N-(2-hydroxypropyl)methacrylamide] (PHPMA) backbone polymer, using a copolymerization of the methacrylate terminal group from the o-nitrobenzyl one to produce an amphiphilic PHPMA-b-poly(DNMA) copolymer. Nanomicelles based on PHPMA-b-poly(DNMA) were developed via the nanoprecipitation method, and further washed by dialysis (Figure 4B) [150].

Jiang et al. developed an illustrative amphiphilic copolymer based on pyrenylmethyl ester (Table 1D) as a hydrophobic segment (PPy) and poly(ethylene oxide) (PEO) as a hydrophilic block. Polymerization of pyrenemethyl methacrylate produced the PPy block, functionalized with the hydroxyl group from PEO to produce the PEO-b-PPy photosensitive amphiphilic polymer. The nanoprecipitation method with THF/H_2_O was employed to produce the micelles based on PEO-b-PPy (Figure 4C) [151]. Arjman et al. synthesized photosensitive polymeric structures, transforming coumarin (Table 1E) to act as a crosslinker between two segments (alkene–coumarin–alkene), polymerizing the alkene segments to self-assemble spheres. The available hydroxyl group of the coumarin was modified by nucleophilic substitution to produce 7-(allyloxy)-4-methyl coumarin double-linked with benzophenone, obtaining two alkene terminals. The alkene groups on each side were then polymerized with the azobisisobutyronitrile technique, forming the photosensitive polymeric spheres (Figure 4D) [152].

### 3.3. Photostimulation Mechanisms for Cargo Delivery

UV and NIR photostimulation of PMNs affect their ionic force, size, shape, stereochemistry, and physicochemical properties. Such stimuli result in photoisomerization and photocleavage of PAMs, leading to cargo delivery (Figure 1(Ad)) [56]. UV light triggers photoisomerization from the hydrophobic character of the *trans* isomer to the hydrophilic or less hydrophobic *cis* isomer of the UV photosensitive PAMs (Table 1A,B), disrupting the hydrophobic/hydrophilic balance of PMNs, and delivering the cargo [153]. For example, after UV irradiation, Roche et al. delivered hydrophobic Nile red and hydrophilic rhodamine B from PEG_n_-b-PCAzo_m_ nanomicelles and nanopolymersomes, respectively. The azobenzene photoisomerization (Table 1A) affected the micelles’ core polarity and the polymersomes’ interface, destabilizing the polarity balance and changing their size [154]. Likewise, Razavi et al. released doxorubicin encapsulated within photoisomerizable spiropyran (Table 1B)-linked poly (N-isopropylacrylamide)-b-poly (methyl methacrylate) nanomicelles via UV irradiation. Photoisomerization of spiropyran chain end groups increased the polarity of the nanomicelles, enabling the cargo release [148].

UV and NIR irradiation cause photostimulation, breaking the specific covalent bond adjacent to UV–NIR PAMs (Table 1C–F) in the PMNs irreversibly, disrupting the hydrophobic/hydrophilic balance and, thus, causing the PMNs’ destruction, and thereby the cargo release [116]. For example, Huo et al. released Nile red as a drug model, using photosensitive polymeric nanomicelles based on PEG-light-responsive-poly(2-nitrobenzylmethacrylate), because the UV irradiation broke the o-nitrobenzyl apart (Table 1C), and the polymeric block changed from hydrophobic to hydrophilic [155]. Similarly, Zhang et al. delivered doxorubicin loaded in poly(ε-caprolactone)-acetal-nitrobenzyl ester-PEG nanocarriers. UV irradiation broke the hydrophobic and hydrophilic polymeric blocks via the photocleavage mechanism, as with the o-nitrobenzyl [156]. Likewise, Jiang et al. encapsulated Nile red into micelles and released it under UV-light irradiation, using a block copolymer synthesized from PEO-b-PPy, based on spiropyran as a hydrophobic segment (Table 1D). The photostimulation ruptured the hydrophobic segment and disrupted the micelles, thus releasing the cargo [151].

Similarly, Razavi et al. developed micelles using a photo-crosslinking strategy for the release of doxorubicin. The micelles were based on poly(amidoamine)-g-[polystyrene-b-(2-(dimethylamino)ethyl methacrylate-co-11-((4-methyl-2-oxo-2H-chromen-7-yl)oxy)undecyl acrylate)] and, after UV irradiation, the crosslinkable coumarin (Table 1E) was separated from the micelle structure, illustrating another photocleavable strategy for cargo release [157].

### 3.4. Characterization of Photosensitive Micro/Nanocarriers

Different alternatives exist for the assembly and characterization of PMNs of different composition, structural configuration, morphology, size, polar and hydrophobic cargo loading capacity, photostimulation wavelengths, and phototriggering mechanisms [158]. Hence, a complete interrogation of PMNs via microscopy and spectroscopy techniques, optical systems, and analytical methods is necessary in order to establish their morphological and physicochemical characterization, LC and EE, cargo photorelease (Table 2), classification, proof of new concepts, and suitable applications.

#### 3.4.1. Cargo Loading Capacity and Encapsulation Efficiency

Cargo has various physicochemical characteristics, and its interaction with PMNs can be mediated by polar tendency affinity, covalent linking, and adsorption, among others, affecting their LC [197,198]. Understanding these cargo–PMN interactions can help in choosing a proper cargo loading method [199] and PMN type [200], and develop new cargo loading approaches with optimal LC and EE (Figure 1(Ac)) [201]. Cargo encapsulation is achieved either by solubilizing the cargo in the hydrophobic or hydrophilic solvent in the PMNs’ formation process [202], or by dispersing the cargo with the PMNs under constant stirring while loading the cargo via the diffusion mechanism [147]. After both processes, the loaded PMNs are washed to remove the free cargo. Whereas LC is the amount of cargo loaded per weight unit of the micro/nanocarriers, the EE is the weight fraction of the cargo payload in the micro/nanocarriers [203]. The gravimetric method is often used to estimate the LC and EE, utilizing the mass ratio between the weight of cargo in the PMNs and the weight of the PMN-loaded cargo (Equation (1)), and the mass ratio between the cargo weight in the PMNs and the cargo input weight (Equation (2)), respectively [204].
LC (%) = (weight of cargo in the PMNs/weight of PMN-loaded cargo) × 100 (1)
EE (%) = (weight of cargo in the PMNs/weight of cargo infeed) × 100 (2)

For example, Zhou et al. encapsulated paclitaxel (PTX) while forming nanomicelles from poly[(N-isopropylacrylamide-co-N,Ndimethylacrylamide)-block-propyleneacylalkyl-4-azobenzoate] by solubilizing the PTX and the copolymer in the organic solvent (THF). The nanomicelles were formed by nanoprecipitation, followed by solvent evaporation, and the LC and EE were estimated to be 32.5 and 83.7%, respectively, by gravimetry [202]. Kim et al. synthesized light-stimulated nanomicelles based on PEG-block-poly-l-lysine modified with a light-responsive group to encapsulate doxorubicin in a nanoprecipitation process, determining an LC and EE of 13.5 and 90.3%, respectively, as measured by spectrophotometry [189]. Molla et al. encapsulated hydrophobic Dil molecules and hydrophilic R6G molecules into nanopolymersomes based on PEG-Azo-PLA, using the diffusion mechanisms, and obtaining an LC and EE of 29 and 40%, respectively, for Dil, and 2 and 13%, respectively, for R6G, via the gravimetric method [147].

#### 3.4.2. Cargo Photorelease

Cargo concentration curves can be plotted using HPLC and spectrophotometry quantification methods, or inferred directly from the other mentioned methods (Table 2). Once the cargo is loaded, different UV or NIR irradiation times are evaluated to determine the accelerated cargo release kinetics, because the irradiation activates the PAMs’ configurational changes, destabilizing or disrupting the PMNs, which is easily tracked by spectrophotometry [205,206,207]. For example, Molla et al. developed PEG-Azo-PLA nanopolymersomes to load hydrophobic Dil molecules. Cargo release time was determined to be 240 min of UV irradiation by tracking the cargo concentration via the UV–Vis absorbance technique over time [147].

## 4. Photosensitive Polymeric Nanomicelles

Photosensitive polymeric nanomicelles (PNMs) are composed of hydrophilic and hydrophobic segments, whose PAMs have apolar characteristics, including azobenzene, spiropyran, and ortho-nitrobenzene, among others, which change their resonant structure via UV-light photostimulation [208]. The difference in the two segments’ hydrophobicity makes the PNMs form micelles with the photosensitive segment in the inner core (Figure 1(Aa,b)) [209]. PNMs enable spatiotemporal control over the cargo release [210]. The photoisomerizable molecule changes with exposure to UV light, to a more polar and less stable form, destabilizing the hydrophobic/hydrophilic balance of the PNMs, and delivering the hydrophobic cargo [142]. Figure 5 illustrates the chemical composition, TEM micrography, photostimulation, and phototriggered cargo release characterizations of representative examples of photosensitive nanomicelles. It can be seen how changes in the absorption spectra of PNMs in response to different light-irradiation times characterize the photostimulation and cargo delivery kinetics. The phototriggering is evident from TEM images after photostimulation and changes in the size distribution and cargo delivery kinetics. Illustrative examples of photosensitive nanomicelles are presented below.

Wu et al. designed a typical UV-light-photosensitive polymer based on azo polyelectrolytes (PPAPE), using poly(acryloyl) chloride as the polymer backbone, azobenzene as the hydrophobic segment, and carboxylic groups of the polymer as the hydrophilic segment. The azobenzene molecule in the polymer was isomerized after 10 s of UV irradiation [214]. Subsequently, Nan Li et al. developed demonstrative PNMs based on PPAPE via nanoprecipitation with THF and H_2_O solvents, forming the PNMs after slow THF evaporation. Photoinduced destabilization was reported after 30 s of UV irradiation at 365 nm, validating the concept of cargo photodelivery (Figure 5A) [211]. Xu et al. reported synthesizing a novel random copolymer based on methacrylate isobutyl polyhedral oligomeric silsesquioxane and azobenzene derivatives. PNMs were developed utilizing the photosensitive polymers in an aqueous solution, and Nile red was loaded to test the drug encapsulation concept. The azobenzene group changed the diameter of the nanomicelles from 121.1 to 107.0 nm after 5 min of UV irradiation, photoreleasing the hydrophobic molecules (Figure 5B) [180].

Similarly, photosensitive micro/nanomicelles (PMNMs) based on hydrophobic drug-loaded photoactivatable emulsifiers have been assembled for cargo administration. For instance, Zhao et al. reported curcumin-loaded PMNMs using β-cyclodextrin-grafted alginate and azobenzene derived from amphiphilic Pickering emulsion. The hydrophobic/hydrophilic balance changed in the PMNMs because the external UV irradiation photoisomerized the azobenzene molecules. The change in the amphiphilic balance destabilized and demulsified the micelles, causing the drug delivery (Figure 5C) [212].

Li et al. developed photocleavable PNMs for a drug/gene/protein delivery approach based on poly(o-nitrobenzyloxycarbonyl-L-lysine)-b-PEO as a new polypeptide copolymer via physical crosslinking and self-assembly, employing methanol and aqueous solutions for the formation process. This work aimed to provide a photostimulus platform based on the construction of polypeptide copolymer hydrogels, in order to study the photosensitive mechanism. After 30 min of UV irradiation with 100 mW cm^−2^ photocleavage intensity, the 40% ortho-nitrobenzyl groups of the PNMs reduced in size rapidly, from 149 to ~58 nm, demonstrating their worth as a PNM-based candidate for cargo delivery (Figure 5D) [213]. In the same context, Alemayehu et al. developed photosensitive supramolecular nanomicelles using complementary adenine (A) and uracil (U) groups and poly(propylene glycol) (PPG) to develop two polymers (A-PPG and U-PPG) for photodelivery of doxorubicin into cervical HeLa cancer cells. In this case, the temperature of the PAMs increased from 37 to 42 °C under UV irradiation, destabilizing the micelles and causing the drug release [204]. Gebeyehu et al. designed a photo- and thermoresponsive uracil-based polymer to produce PNMs with a size of 193.0 nm via supramolecular interactions. PNMs showed UV-sensitive photodimerization, low cytotoxicity in MCF-7 cells, and tunable doxorubicin-loading capacity. UV irradiation at 254 nm augmented the temperature from 25 to 40 °C, along with the hydrophilic–hydrophobic phase transition, ensuring that the hydrophobic drug was delivered [209].

UV- and NIR-PAMs are used to develop PNMs for controlling the cargo release kinetics, incorporating side groups into the hydrophobic core-forming block, including at the two-segment junctions [215,216]. UV and NIR irradiation destabilize the PNMs, releasing the cargo [217]. For example, Cao et al. developed new NIR-PNMs based on o-nitrobenzyl PAMs and the N-succinyl chitosan polymer. One hour of NIR irradiation at 765 nm dissociated the PNMs for a photocleavage reaction, and the cypate dye as the drug model was released in deep tissues (Figure 5E) [149].

## 5. Photo-Stimulated Organic and Polymeric Nanocarriers for Cargo Delivery

Photosensitive organic and polymeric nanocarriers—such as liposomes, polymeric nanospheres or nanoconjugates, and nanopolymersomes—are characterized by having PAMs in their structure for cargo release with spatiotemporal control [218], and can be differentiated by their composition and shape, as mentioned above [219]. Figure 6 shows the chemical structure, TEM images, photostimulation, and phototriggering characterization of some photosensitive nanocarriers. Photostimulation and phototriggering were characterized by absorption spectra based on UV–NIR irradiation time, TEM micrographs, variation in the PMNs’ size, and the cargo release kinetics. Illustrative examples of photosensitive organic and polymeric nanocarriers for cargo delivery are presented below.

### 5.1. Photosensitive Liposomes and Polymeric Nanoconjugates

Incorporating PAMs into the structure of liposomes is a strategy to decrease cargo release time, using the photocleavage and photoisomerization mechanisms mentioned above [223]. Encapsulating or functionalizing the PAMs in the hydrophobic bilayer membrane or the hydrophobic head of the phospholipid may destabilize the bilayer membrane via photoactivation [224,225]. For example, Lui et al. developed light-photoisomerizable nanoliposomes embedded with azobenzene for doxorubicin delivery, encapsulating the PAMs in the hydrophobic interface. When the azobenzene molecule was photostimulated, a change from *trans* to *cis* occurred, generating a change in the structure’s hydrophilic character that destabilized the liposome solubility balance and triggered the cargo release [226]. In contrast, Li et al. designed a light-cleavable lipidoid nanoparticle conjugated with nitrobenzene in the phospholipid head and UV light to break the covalent bond, shedding the hydrophobic block, disaggregating the liposome and, thus, releasing the cargo [227].

The photoactivation of UCNPs covered or linked with polymers and PAMs is another strategy for cargo delivery (Figure 1(Ab)) [220]. UCNPs are inorganic nanoparticles based on lanthanides that can emit more energetic UV light under specific NIR light exposure, while the emitted light activates the PAMs for the cargo photorelease [228]. For example, Li et al. developed NIR-photosensitive nanospheres based on UCNPs covered with PEG and functionalized with Cys-Arg-Gly-Asp peptide and loaded nitrobenzene molecules for photodelivery into human mesenchymal stem cells. The peptide interacted with the specific cells’ bioreceptors, enabling the conjugate cell uptake. The UCNPs emitted UV light via NIR irradiation at 980 nm, causing photocleavage of the nitrobenzene molecule and delivery of the model drug into cells, enhancing neocartilage formation in vivo (Figure 6A) [220]. 

Zhao et al. designed UCNP-based NIR-photosensitive polymeric nanospheres via a layer-by-layer co-assembly methodology for photodelivery of doxorubicin into U87-MG tumor cells. The UCNPs were the nuclei of the nanoconjugate; different layers of azobenzene-functionalized polymer covered the nucleus, while the drug was loaded in between these layers. The nanospheres were irradiated with NIR light at 980 nm, and the UCNPs inside emitted UV light, activating the azobenzene molecules present in each layer. The photoisomerization separated each layer, disrupting the nanoconjugate and delivering the drug into cancer cells (Figure 6B) [221]. On the other hand, Zhao et al. reported O_2_-self-sufficient photosensitive polymeric nanovesicles (water–oil–water phase structure) based on PEG-poly(ε-caprolactone). The aqueous phase in the core encapsulated perfluorooctyl bromide, the oil phase loaded the IR780 photosensitizer, and the shell had a functionalized Cys-Arg-Gly-Asp-Lys peptide for specific cellular delivery. NIR irradiation activated the perfluorooctyl bromide, supplying O_2_, and the polymeric nanospheres released the photosensitizer. The combination of O_2_ and the photosensitizer improved the limited effectiveness of photodynamic therapy due to hypoxic microenvironments when tested in the MDA-MB-231 cancer cell line. Remarkably, the chain reaction demonstrated spatiotemporal control against reactive oxygen species (ROS) [229].

### 5.2. Photosensitive Nanopolymersomes

The functionalization of hydrophobic PAMs in the chemical structure of the amphiphilic polymers for the formation of nanopolymersomes is a strategy to improve the cargo release kinetics [230]. PAMs can link the hydrophobic and hydrophilic segments of the polymer (Figure 3B) or the hydrophobic segments of the polymersome (Figure 6C). UV–NIR irradiation photoactivates the PAMs, destabilizing the hydrophobic/hydrophilic balance via photoisomerization and photocleavage mechanisms for spatiotemporal control of cargo delivery [231]. For example, Molla et al. developed a new light-induced interfacial layer from an oil–azobenzene–water nanopolymersome of 100 nm in diameter, encapsulating rhodamine 6G dye (hydrophilic molecule) and Dil dye (hydrophobic molecule). UV irradiation at 360 nm photoisomerized the azobenzene molecule, destabilizing the hydrophobic segment and releasing the cargo [147].

Similarly, Hou et al. designed P-NPs of 150 nm in diameter based on poly (o-nitrobenzyl acrylate) hydrophobic and poly(N,N′-dimethylacrylamine) hydrophilic blocks for hydrophobic doxorubicin encapsulation and photorelease, with potential use in cancer therapy. UV irradiation at 365 nm caused the photocleavage reaction of o-nitrobenzyl hydrophobic groups and disrupted the polymersomes for the drug release (Figure 6C) [140].

Zhou et al. reported lipid-like amphiphilic polymersomes of 125 nm in size for photodelivery of doxorubicin into HeLa cells. The ammonium moiety was the hydrophilic segment of the P-NPs that provided affinity toward overexpressed folate receptors targeting cancer cells. The o-nitrobenzyl was the hydrophobic segment, whose photocleavage was used for drug delivery. The cellular assay showed the affinity of the P-NPs for the cancer cells, along with the intracellular drug delivery triggered by UV irradiation at 365 nm (Figure 6D) [220].

Tang et al. developed NIR-P-NPs for photodynamic therapy and chemotherapy. Poly (propylene sulfide)-b-PEG was the base of the polymersome, the hydrophobic photosensitizer zinc phthalocyanine was loaded into the shell, and the hydrophilic doxorubicin hydrochloride was in the core. The NIR-P-NPs produced singlet oxygen oxides and sulfur molecules on the polymersome via NIR irradiation at 660 nm, causing their disruption and delivering the doxorubicin. The inhibition of A375 tumor growth using dual therapy proved high efficiency for cancer treatments [232].

## 6. Functionalization of Phototriggered Nanocarriers for Targeting Cells

Functionalization of nanocarriers with targeting biomolecules can increase the cargo concentration in specific cells with fewer doses [233]. The specific targeting biomolecules interact with the cellular membrane receptors for the nanocarriers’ uptake into cells (Figure 1(Ac)) [234]. This process is commonly mediated by endocytosis for reduced side effects and optimized cargo delivery, impacting on the effectiveness and efficiency of the therapeutic regimens [235,236]. Phototriggering nanocarriers provide not only spatiotemporal control of cargo delivery [237], activating and controlling the kinetics of the related processes [238], but also reasonable control for fluorescence imaging [239] and photodynamic therapy [240], among other applications (Figure 1(Ae)) [241]. Nanocarriers are synthesized with free functional groups with a positive or negative net charge, porosity, and adsorption characteristics on their surface for functionalization, targeting biomolecules via covalent linking, adsorption, and π–π interaction strategies, promoting specific interactions with the target cells [242,243,244]. Fernández and Orozco summarized different chemistries for functionalizing photosensitive nanocarriers in a recent review [245].

Mena-Giraldo et al. developed a cell-specific nanobioconjugate for cargo photorelease in cardiac cells. Photosensitive chitosan matrix nanoparticles based on azobenzene PAMs co-encapsulated Nile red and dofetilide as drug models. In an amidation reaction, free amines of a cardiac-targeting peptide were functionalized with the carboxylic groups on the nanoparticles’ surface. The peptide caused a specific cellular uptake of the nanobioconjugate via interaction with the cardiomyocytes. UV irradiation at 365 nm triggered the photorelease of the drug into the cells. The azobenzene hydrophobic segment was photoisomerized, changing its polarity and destabilizing the hydrophobic/hydrophilic balance of the nanobioconjugate, thereby providing cell-specific and spatial activation control of the drug release (Figure 7A) [146].

Similarly, Hong et al. synthesized phospholipid-PEG fluorophore nanoparticles with cellular specificity for NIR fluorescence imaging in vivo. The polymer core encapsulated a fluorophore, the surface linked to PEG, and their terminal amine linked to an anti-EGFR antibody. Antibody–bioreceptor interaction between the nanoconjugate and the MDA-MB-468 cells allowed for a specific adherence. NIR ≥ 1000 nm activated the cell fluorescence of the nanoconjugate on the surface of the cells, showing a specific and ultrafast strategy for in vivo fluorescence imaging of cancer cells (Figure 7B) [246]. Zhang et al. designed dimer-targeting, peptide-mediated UCNPs for specific drug delivery and breast cancer therapy. The UCNPs—based on yttrium(III) chloride (YCl_3_), ytterbium(III) trichloride (YbCl_3_), and thulium(III) chloride (TmCl_3_)—were synthesized via the nanoprecipitation method, coated with a mesoporous silica shell, and loaded with epirubicin as a drug model. The photosensitive ortho-nitrobenzene β-CD-based capping structure acted as a plug for the pores, preventing cargo delivery. The adamantane-dimer-targeting peptide was linked to another end of the ortho-nitrobenzene structure, interacting with the overexpressed receptors in the breast cancer cells for the nanoconjugate-specific uptake. NIR irradiation at 980 nm excited the UCNPs for UV light conversion, and the emitted UV light caused the photocleavage of the ortho-nitrobenzene, releasing the pores and delivering the drug into the cancer cells (Figure 7C) [247].

Jadia et al. synthesized some transferrin-targeting polymeric nanoparticles for phototriggered treatment of breast cancer. PLGA-PEG-based nanocarriers encapsulated verteporfin as a photosensitizer, while the surface of the nanoparticles was functionalized with transferrin-receptor-targeting peptides for cellular uptake via their interaction with the transferrin receptor overexpressed in the cellular membrane. NIR irradiation at 690 nm excited the photosensitizer for photodynamic therapy of cancer cells (Figure 7D) [248]. Yang et al. modified liposomes with a photosensitive cell-penetrating peptide and asparagine–glycine–arginine peptide ligand for targeted delivery of siRNA. The liposomes were loaded with the siRNA, and a PEG chain was functionalized with a photolabile-caged cell-penetrating peptide (pcCPP) and asparagine–glycine–arginine molecules. The photosensitive ortho-nitrobenzene group of the modified peptide neutralized the charge of the liposomes, and the peptide interacted with the cellular membrane specifically. NIR irradiation at 740 nm phototriggered a breakage of the covalent bond of the ortho-nitrobenzene, and the charge balance changed for the liposome’s cell uptake. The siRNA was delivered intracellularly after the specific internalization (Figure 7E) [249].

**Figure 7 polymers-13-03920-f007:**
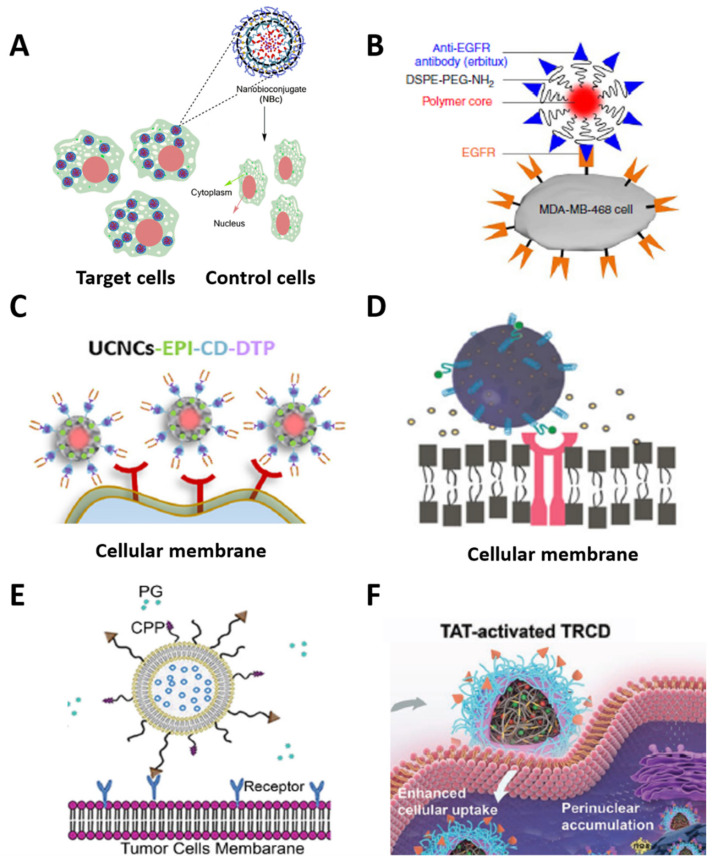
Schematic representation of phototriggered nanocarriers functionalized with cell-targeting biomolecules for specific cargo delivery and imaging: (**A**) cell-specific photosensitive nanobioconjugates for cargo delivery in cardiac cells, reprinted from [146], copyright 2020, Springer Nature; (**B**) phospholipid-PEG fluorophore nanoparticles with cellular specificity for NIR fluorescence in in vivo imaging, reprinted from [246], copyright 2014, Springer Nature; (**C**) dimer-targeting peptide-mediated UCNPs for specific drug delivery and enhanced breast cancer therapy, reprinted from [247], copyright 2021, Elsevier; (**D**) verteporfin-encapsulated PLGA-PEG nanocarriers for cell-specific photodynamic therapy, reprinted from [248], copyright 2018, John Wiley and Sons; (**E**) photosensitive liposomes for cellular targeting and siRNA delivery, reprinted with permission from [249], copyright 2015, Elsevier; and (**F**) dual pH/photosensitive polymeric nanocarriers for specific drug delivery into cell nuclei and improved cancer therapy, reprinted with permission from [250], copyright 2019 John Wiley and Sons. Images modified with permission.

Cao et al. developed dual pH/photosensitive polymeric nanocarriers for specific drug delivery into cell nuclei and enhanced cancer therapy. Thioketal-crosslinked polyphosphoester-based nanoparticles and doxorubicin-loaded hyperbranched nanoparticles (^D^TRCD) were synthesized via the crosslinking methodology and linked to a pH-sensitive peptide. The peptide permitted specific cellular uptake in tumor tissues. NIR irradiation at 660 nm activated the ^D^TRCD for ROS generation, causing cellular membrane disruption, nucleus internalization, and drug release (Figure 7F) [250].

Overall, this section discussed different PAMs to develop UV–NIR PMNs based on photoisomerization and photocleavage mechanisms for accelerated cargo photodelivery in biomedical interventions, highlighting the versatility of the phototriggered nanocarriers for specific cellular uptake, as summarized in Table 3.

## 7. Polymeric Micro/Nanomotors for Cargo Transport

### 7.1. Definition and Classification

Motion is the translation or rotation of a body from one reference point to another [251], tracking a distance in a time, quantified by velocity [252]. A body in a medium without motion has potential or static energy [253], representing a system in equilibrium [254]. Gradients of temperature [255] and concentration [256], along with transfer of momentum [257], electrons [258], protons [259], photons [260], and sound waves [261], alter the body’s systemic equilibrium, transforming the potential energy into a dynamic state with kinetic energy [253]. Micro/nanomotors (MNs) are devices that suffer internal and external fluctuations in fluids by interacting with their environment and external fields, converting their static energy into kinetic energy and, thus, generating mechanical movement [262] that often seeks to accelerate the kinetics of the involved process, improving its efficiency (Figure 1(Bf)) [263].

The MNs are classified by composition, shape, and propulsion mechanisms [264]. The MNs comprise proteins [265], metals [266], hydrogels [267], organic materials [268], and polymers [269] that conform to different shapes, including tubular [270], spiral [271], and Janus particles [272]. The MNs’ variety of compositions and shapes permits high interaction with different cargo types—such as antigens [273], DNA [274], proteins [275], drugs [276], photosensitizers [277], and pollutants [278]—and transport, protection, and delivery of cargo [279] for multiple applications, ranging from cell targeting [280] and imaging [281] to biosensing [282], environmental [283], and biomedical [284] intervention.

The principal propulsion mechanisms are photic [285], magnetic [286], catalytic [287,288], self-electrophoretic [289], self-diffusiophoretic [290], and the Marangoni effect [291], showing the versatility of the MNs in generating motion [292]. In particular, polymeric and catalytic Janus-particle-based MNs consist of an anisotropic particle with two faces with different characteristics so as to generate a specific reaction on one side to produce the motion while performing different tasks (capturing, transporting, (bio)sensing, etc.) on the other side, using the polymeric structure for cargo interaction [293]. Based on the propulsion mechanisms, MNs can be classified as follows:

#### 7.1.1. Propelled by Light

Thermophoresis is the migration of colloidal particles into a solution as a result of heat diffusion [294]. Materials with high conductivity and thermal/photo-response—such as gold—suffer an electronic excitation, manifested in the form of heat and stimulated by light-irradiation [295]. Micromotors based on gold increase the temperature at a specific site, generating a temperature gradient and causing their motion into a fluid [296]. Light-stimulus-based micromotors are attractive as a fuel-free motion alternative for noninvasive intervention and spatiotemporal external control approaches [297], such as cargo transport, environmental intervention, biosensing [298,299], and cancer therapy [300]. For example, Wu et al. reported a fuel-free NIR-driven Janus capsule motor based on silica particles and gold. The anisotropic particle fabrication consisted of a layer-by-layer technique covering silica particles of 1–20 μm in diameter with a gold layer on one side via the sputtering deposition method. The NIR irradiation at 808 nm excited the gold and activated the self-thermophoresis effect, producing motion due to a resultant temperature gradient. The Janus micromotor showed a maximum speed of 42 μm s^−1^ in water and cell culture media, generating a dual mechanism—thermophoretic force for the motion, and dynamic photothermal therapy for cellular apoptosis—improving efficiency and presenting an active strategy for cancer therapy (Figure 8A) [301].

In the same way, Wu et al. reported a polymeric tubular rocket of 10 μm in length and 5 μm in width, based on poly(styrene sulfonic acid) and poly(allylamine hydrochloride), functionalized with gold nanoshells inside the pores of the tubular polymeric structure. NIR irradiation at 780 nm activated the gold nanoparticles for specific site heating, generating a radial temperature gradient. The heat dispersion through the tubular polymer photo-propelled the motor at 160 μm s^−1^ in cell culture media, demonstrating fuel-free and externally controlled propulsion for biomedical applications (Figure 8B) [302].

#### 7.1.2. Propelled by Magnetic Fields and Catalytic Reactions

Electric current, charge, and magnetic materials generate a magnetic field with a vector domain and a perpendicular force that repels or attracts other magnetic materials [303,304]. MNs based on magnetic materials—such as magnetite [305]—use a simple external magnetic field, changing their polarity to high motion control [306]. In the same context, catalytic reactions are strategies for MN motion using chemical and biological reactions that produce gas, protons, and concentration gradients, driving the MNs [279,307]. For example, Janus-MNs use the reaction of platinum (Pt) nanoparticles (NPs) with hydrogen peroxide (H_2_O_2_) to produce oxygen and hydrogen bubbles in the anisotropic particles in order to generate a driving force for the generation of motion [308]. Furthermore, polymeric MNs based on catalytic motion have been developed using polymeric spheres, polymersomes, and polymeric stomatocytes for cargo transport interactions [309].

**Figure 8 polymers-13-03920-f008:**
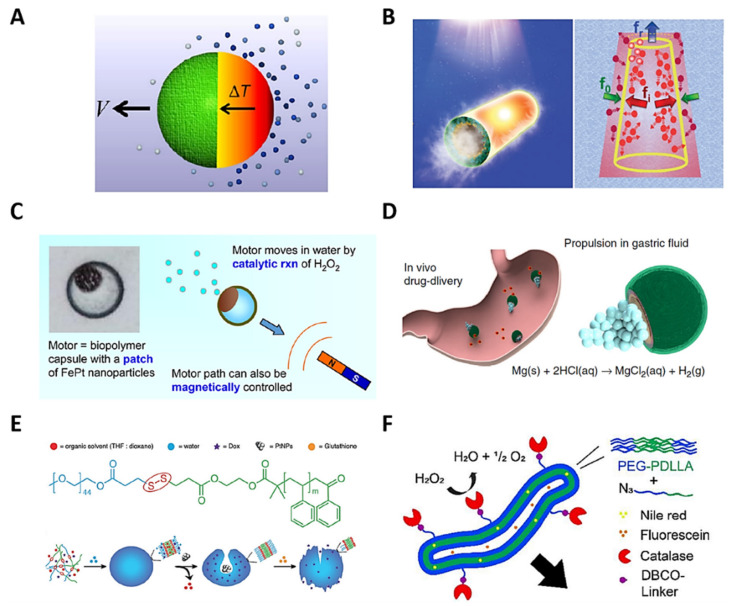
Schematic representation of polymeric micro/nanomotors’ propulsion mechanisms: (**A**) light-propelled JMs, reprinted with permission from [301], copyright 2016, Springer Nature; and (**B**) polymeric tubular rocket, reprinted with permission from [302], copyright 2016, John Wiley and Sons. Magnetic fields and catalytic reactions for propulsion: (**C**) magneto-catalytic polymeric microcapsules, reprinted with permission from [310], copyright 2016, American Chemical Society; (**D**) catalytic polymeric sphere micromotor, reprinted with permission from [311], copyright 2017, Springer Nature; (**E**) catalytic polymeric stomatocyte nanomotors, reprinted with permission from [312], copyright 2017, John Wiley and Sons; and (**F**) enzymatic polymersome nanomotor, reprinted with permission from [313], copyright 2018, Royal Society of Chemistry.

Li et al. developed magnetic, externally controlled and self-propelled JMs for thermo-recognition and erythromycin antibiotic adsorption in water. Porous magnetic microparticles of CoFe_2_O_4_ with an average diameter of 50 μm were synthesized and covered with Mn_3_O_4_ using a heat treatment methodology. The microparticles were dispersed in polyvinylpyrrolidone on a glass slide and coated with a hydrogel-like stimulus-sensitive polymer based on poly (N-isopropylacrylamide). They were then removed from the substrate and exposed to the CoFe_2_O_4_/Mn_3_O_4_ core for anisotropic particle formation, permitting the reaction between Mn_3_O_4_ and H_2_O_2_ to produce water and oxygen bubbles for catalytic propulsion at a velocity of 161 ± 9 μm s^−1^. The CoFe_2_O_4_ had magnetic properties as a result of the presence of Fe, providing external magnetic control. The sensitive polymer of the JMs adsorbed erythromycin upon reaching a critical solution temperature (33.9 °C), and delivered it with a lower temperature, accelerating the interaction with the medium via micromotor motion. This characteristic, combined with the magnetic control and the catalytic self-propulsion, showed the potential for drug delivery in biomedical applications and antibiotic adsorption in water purification alternatives [314].

Lu et al. reported polymeric Janus microcapsules with magneto-catalytic motion, employing a crosslinking reaction between chitosan and glutaraldehyde. A microfluidic system was used to form chitosan droplets with encapsulated iron (Fe)–Pt NPs and an external magnetic field to make Fe–Pt NPs aggregate in the corners of the chitosan droplets. At the same time, the crosslinking reaction generated anisotropic and solid capsules of ~150 μm in diameter. The interaction between H_2_O_2_ as the fuel and Fe–Pt NPs produced catalytic motion, with oxygen bubbles as the driving force. H_2_O_2_ concentrations of 3–16 wt% were evaluated, producing motor speeds of 200–2400 μm s^−1^, respectively. A magnetic field provided magnetic motion due to the presence of iron nanoparticles. Overall, the chitosan micromotors demonstrated cargo transportability, adhering to other chitosan microparticles, and delivering the cargo along a specific path toward a controlled destination by changing direction via a magnetic field. Furthermore, a model dye was loaded and delivered by chitosan micromotors via a diffusion mechanism, demonstrating the potential usefulness of soft micromotors for the targeted delivery of cargo (Figure 8C) [310].

Esteban-Fernández de Ávila et al. developed a catalytic Janus polymeric sphere micromotor based on magnesium (Mg), titanium dioxide (TiO_2_), PLGA, and chitosan for drug delivery and treatment of a stomach infection. The micromotors, with an average size of ~20 μm, were assembled via the layer-by-layer strategy, implementing Mg nanoparticles in the core, and covering them with TiO_2_ via the atomic layer deposition technique to stabilize the micromotor core. The drug-loaded PLGA and chitosan-coated layer were assembled via polymer deposition, where chitosan showed biocompatibility and affinity with the stomach tissue. The motion, with an average speed of ~120 μm s^−1^, was caused by the catalytic reaction between Mg nanoparticles and hydrogen chloride, simulating gastric fluid of pH ~1.3 to generate hydrogen bubbles. An antibiotic drug was delivered to achieve a significant bacterial burden reduction in the murine stomach (Figure 8D) [311].

Tu et al. reported polymeric Janus stomatocyte nanomotors based on redox-sensitive PEG-S-S-polystyrene with Pt nanoparticles in the core for catalytic motion and doxorubicin delivery via H_2_O_2_ production in the biotic-reducing environments of the cancer cells. PEG-S-S-polystyrene was synthesized via styrene polymerization and quenched with a PEG-SS-Br macroinitiator to obtain an amphiphilic polymer with PEG and polystyrene as the hydrophilic and hydrophobic segments, respectively. The nanoprecipitation method and vigorous dialysis were used to form polymeric stomatocyte nanomotors of 345 nm in diameter and Pt NPs in the core, solubilizing PEG-S-S-polystyrene and Pt NPs in the apolar THF/dioxane and polar water solvents, respectively. H_2_O_2_ present in the cancer cells acted as a biofuel, producing H_2_O_2_ concentration gradients of 5 nmol per 10^4^ cells h^−1^. The catalytic reaction between Pt nanomotors and H_2_O_2_ produced oxygen bubbles that generated motion with a velocity of ~35 μm s^−1^ in the direction of a high-H_2_O_2_-concentration cellular area. Doxorubicin was released in the cancer cells because the copolymer’s S-S bond was disrupted by the glutathione endogenous reducing agent present under biological conditions, breaking apart the hydrophobic and hydrophilic stomatocyte segments. These nanomotors were a new concept for motion and cargo transport and delivery for biomedical applications in the cellular environment (Figure 8E) [312].

#### 7.1.3. Propelled by Enzymes

Enzymes are proteins that can convert a substrate into a product via catalytic reactions, having high applicability in biological motion at the cellular level [315]. Enzyme-based motors, taking advantage of this catalytic activity, generate energy and produce motion from hydrolysis, among other reactions [316]. Enzymes exhibit biocompatibility and biodegradability, suitable as motion sources in biomedical and biological approaches [317]. For example, Toebes et al. designed a tubular polymersome nanomotor based on catalase-functionalized PEG-b-poly(D, L-lactide) for catalytic motion and amphiphilic drug delivery. The polymersomes were 90 nm in diameter, self-assembled via the nanoprecipitation method, using THF and dioxane in continuous dialysis. A sodium chloride solution transformed the circular polymersomes to tubular polymersomes of 300 nm in size, via dialysis. Finally, the carbodiimide esterification method was used to link catalase to the tubular polymersomes’ surface. Catalase catalytically reacted with H_2_O_2_ to produce oxygen bubbles, and the motion was achieved because the velocity vectors did not cancel one another out, generating a driving force in a resultant direction. Nile Red and fluorescein were encapsulated in the nanomotor as hydrophobic and hydrophilic drug delivery models, respectively (Figure 8F) [313].

Chen et al. reported protein- and phospholipid-based emulsions, composed of bovine serum albumin (BSA)-, dimyristoyl-sn-glycerol-3-phosphocholine (DMPC)-, and polyvinyl alcohol (PVA)-stabilized Janus surfactant droplets, using lipase and urease for enzymatic motion. The microemulsion technique produced BSA- and DMPC-PVA-based droplets of 22–80 µm and 22 µm in diameter, respectively. Rhodamine B isothiocyanate (RBITC) and coumarin stain agents enabled green and red fluorescence visualization for morphology and tracking characterization. Tributyrin and triacetin were loaded as proof-of-concept cargos, stabilizing the droplets, with 70% of cargo loaded via the fusion mechanism. Janus structures were fabricated to control the fusion frequency between both droplets, showing a high control to produce asymmetric morphology.

Similarly, urease was used as a surfactant and stabilized with oleic acid to form a droplet; at the same time, the droplet was merged with the urease emulsion, forming a Janus-DMPC-urease emulsion. Janus droplets in a urea–aqueous solution produced motion with a velocity from 2 to 41 µm s^−1^ due to the hydrolysis of urea by urease contained in the emulsion, producing a proton gradient. In a similar manner, lipase-based Janus droplets were developed using the fusion between lipase- and DMPC-triacetin droplets, producing a catalytic motion because the lipase hydrolyzed triacetin into glycerol and acetic acid. New Janus droplet formation was demonstrated, taking advantage of the fusion mechanism and the enzyme interaction for motion, stimulating the knowledge of biophysical behaviors with lipid and protein droplets [318].

Wang et al. developed self-fueled lipase-active oil droplets with a triglyceride substrate and polydimethylsiloxane (PDMS) oil mixture for enzyme-induced motion. Tributyrin and PDMS with lipase solution produced a water/oil dispersion with 10 of 60 µm of oil droplets. RBITC-labelled lipase and oil-soluble green fluorescent dye were employed to characterize the cargo, its distribution, and its morphology via confocal fluorescence microscopy, showing membrane thickness of 7.46 nm, consistent with lipase’s molecular size. The motion was produced by the protocell buoyancy mechanism between lipase-coated lipid PDMS droplets and the loaded tributyrin substrate when the temperature was increased to above 17 °C, producing glycerol and butyric acid. New protocell strategies for the specific chemical and temperature gradients can generate motility- and buoyancy-based physical sorting [319].

#### 7.1.4. Propelled by Electrophoresis, Diffusiophoresis, and the Marangoni Effect

Self-electrophoresis motion consists of electrochemical fuel decomposition, generating proton gradients and electric fields in the anodic and cathodic ends of the MNs [320]. For example, Wang et al. designed bimetallic nanorods for self-electrophoretic motion using H_2_O_2_ as a fuel (Figure 9A) [321]. Nanorods with a cylindrical geometry of 3 μm in length and a 150 nm radius were synthesized via a metal electrodeposition methodology, assisted by an alumina membrane template. The nanorods decomposed the H_2_O_2_, consuming the protons and generating a proton gradient and an electric field, in order to produce motion at 21 ± 4 μm s^−1^, used in a collective phenomenon for catalytic motor propulsion, similar to chemotaxis [322]. Instead, an ionic chemical reaction on the surface produces the Janus MNs’ motion for diffusiophoresis, generating an electric field for electrokinetic flows (Figure 9B) [323]. As another illustration, Zhou et al. synthesized JMs based on dielectric AgCl to demonstrate ionic self-diffusiophoresis. Poly (methyl methacrylate) microspheres of 2.5 μm in diameter were thermally covered with a 50 nm layer of silver by an electron-beam evaporator. The microspheres were submerged in an FeCl_3_ solution to produce a thin layer of AgCl and obtain the characteristic Janus shape. UV light or high-power visible light irradiation activated the JMs, producing faster H^+^ and slower Cl^–^ for the electric motion in one direction, and showing negative gravitaxis under illumination, with the potential for 3D cargo transport and sensing applications [324]. 

The Marangoni effect produces milli/micromotor motion by transferring mass in the liquid–liquid interface from a surface tension gradient (Figure 9C) [325]. For illustration, Orozco et al. developed an enzyme-loaded tubular motor self-propelled by the Marangoni effect for pollutant degradation. The tubular motor generated the motion in the medium–surfactant interface, changing the surface tension of the tubular motor while releasing the enzyme. This motion accelerated the pollutant degradation via enzyme delivery, presenting a strategy for enhanced environmental applications [326].

## 8. Some Applications of Light-Stimulated Polymeric Micro/Nanomotors

### 8.1. Propulsion, Cargo Transport, and Drug Delivery

Polymeric micro/nanomotors are composed of polymeric structures that allow for malleability [327], biocompatibility [328], biodegradability [329], high surface area [330], adsorption [331] and absorption [332] ability, cargo transport [333], and reproducibility [334]. Furthermore, polymeric micro/nanomotors containing PAMs can enable spatiotemporal control [335] for different tasks, such as directed propulsion [336] or cargo transport and delivery [337]. The PAMs, under light-irradiation, change their physicochemical characteristics via the mechanisms explained above, interacting with the medium and producing a photomotion [338]. Fuel-free propulsion is an approach in advancing polymeric MNs to deal with high toxicity or limited presence of fuels [339]. This technology enables unlimited fuel and external control, and opens the path for new biomedical applications such as drug delivery, self-adaptive photocatalysts, and micro/nanorobotic devices [340], as depicted in Figure 1(Bg) of this manuscript.

Abid et al. designed azobenzene-coated crosslinked polystyrene nanoparticles of 16 nm in diameter, and the azobenzene PAMs were functionalized on the spheric polymeric surface. The alternation between UV light and visible irradiation generated the motion of the nanomotor, because the azobenzene molecule underwent *trans*–*cis* photoisomerization under UV irradiation and *cis*–*trans* under visible light. This constant change in configuration altered the stereochemistry and polarity of the nanomotor’s surface—only on the side where it was exposed—producing sufficient mechanical work to generate motion with a velocity of around 15 μm s^−1^, constituting a photoaddressable nanodevice approach to light-stimulated propulsion with the potential for nanomechanical drug delivery (Figure 10A) [341]. Similarly, Li et al. developed light-motion-controlled polymeric micromotors based on a photochromic hyperbranched polymer, functionalizing spiropyran PAMs on the available functional groups from the polymer. Colloidal particles were prepared via an emulsion technique, using H_2_O/dimethyl sulfoxide to obtain nanoparticles of 400 nm in diameter, but agglomerated together to form microconjugates of 2.1 μm in size, such that the spiropyran groups were on their surface and had an adequate size for interaction with light. UV irradiation photoisomerized the SP to convert it into a nonpolar zwitterionic merocyanine form. This produced an interfacial tension gradient on the particle’s surface in the water solution that, under constant UV irradiation, activated the motion with a velocity of 20 μm s^−1^ in the direction of the UV emission. This work showed a photostimulated motion proof-of-concept, opening opportunities in self-adaptive photocatalysis, drug delivery, and micromotor technologies [342].

Nanomotor-based strategies are attractive in drug delivery approaches because they are cargo-loaded nanocarriers with sizes usually lower than 100 nm, facilitating cellular uptake, accelerated kinetics, and fuel-free motion [343]. Non-biodegradable components and fuels limit their translational approach in biomedical interventions [281]. Therefore, developing new nanomotor technologies using biopolymers for dynamic cargo transport, intracellular stimulus-delivery, and fuel-free motion is imperative [344]. With this in mind, Shao et al. designed an NIR-photoactive polymersome nanomotor for the delivery of doxorubicin into HeLa cancer cells, based on biocompatible PEG-b-poly(D, L-lactide) connected with a pH-sensitive imine bond and modified with a hemispherical gold nanocoating. The Janus nanomotors, with a size of around 100 nm, were adequately taken up by cancer cells. NIR irradiation produced a Janus polymersome motion as a result of the thermophoretic effect of the gold material, providing velocities of 6.2 μm s^−1^ and enabling an accelerated intracellular interaction. Changes in pH destabilized the polymersome and released the drug model for efficient cancer intervention (Figure 10B) [345].

**Figure 10 polymers-13-03920-f010:**
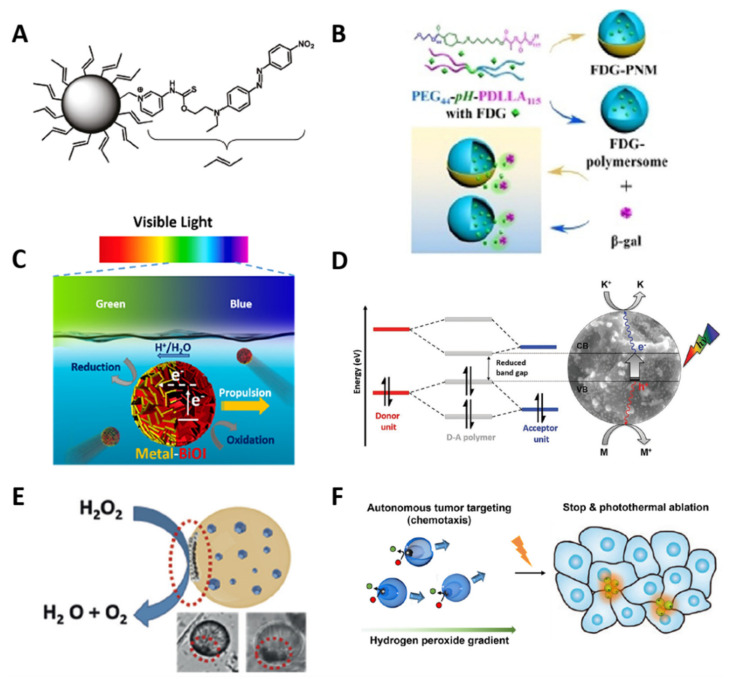
Schematic representation of light-stimulated polymeric micro/nanomotors for propulsion, by (**A**) an azobenzene-coated crosslinked polystyrene nanomotor, reprinted with permission from [341], copyright 2011, American Chemical Society; (**B**) cargo transport—an NIR-photoactive polymersome nanomotor for doxorubicin delivery, reprinted with permission from [345], copyright 2020, John Wiley and Sons; (**C**) visible-light-photosensitive micromotors, reprinted with permission from [346], copyright 2017, American Chemical Society; (**D**) an organic polymer sulfur semiconductor and nitrogen donor/acceptor polymer, reprinted with permission from [347], copyright 2020, John Wiley and Sons; (**E**) light-stimulated and catalytic micromotors enhance the biosensing detection, reprinted with permission from [348], copyright 2017, John Wiley and Sons; and (**F**) NIR thermal and catalytic polymeric nanomotors, reprinted with permission from [349], copyright 2018, American Chemical Society.

Similarly, Wang et al. developed a microfluidic device for polymeric inorganic nanoparticles—a Janus vesicle motor for self-propulsion and cargo release based on amphiphilic PEO-block-poly(styrene) (PS), Pt nanoparticles, and Au nanorods (AuNRs). The THF/water nanoprecipitation-like method, mentioned in the previous sections, was used to form the microfluidic system inside the Janus-like vesicle structure, enabling self-assembly thanks to the amphiphilic polymer characteristics. The vesicle-like polymersomes had the Pt and Au nanoparticles covering a corner of the hydrophobic face, forming a Janus-like vesicle. The device formed 749.2 ± 228.3 to 2634.2 ± 795.3 nm diameter polymersomes with considerable size control on demand. The Pt nanoparticles in the corners of the vesicles reacted with hydrogen peroxide between 0 and 30%, producing oxygen bubbles for propulsion, reaching velocities of 2–90 μm s^−1^. Hydrophilic drug molecules were encapsulated in the polar core of the motor, and NIR irradiation activated the AuNRs, increasing their temperature and destabilizing the vesicle interface, triggering a breakdown of the polymer structure and photodelivering the cargo with spatiotemporal control. The concept of vesicle motion was probed in order to improve the kinetics in drug delivery technologies [350].

Bozuyuk et al. designed magnetic chitosan polymer microswimmers for light-triggered drug release based on double-helical microswimmers and methacrylamide chitosan polymer o-nitrobenzyl PAMs. Methacrylic anhydride reacted with the free amines of the polymer to produce the chitosan-derived PAMs. Superparamagnetic iron oxide nanoparticles (SPIONs) were embedded with the chitosan-derived and 3D-printed PAMs via two-photon direct laser writing (TDLW) to form a magnetic microswimmer of 20 μm length and 6 μm outer diameter. A magnetic field at 10 mT and 4.5 Hz produced a motion with 3.34 ± 0.71 μm s^−1^ of velocity. O-nitrobenzyl PAMs were functionalized on the micromotor surface via the free amines of the chitosan-derived PAMs, and azide-modified doxorubicin was linked to their other side. Thirty minutes of UV light at 365 nm photocleaved the PAMs on the micromotors, releasing 60% of the doxorubicin. Magnetic motion and anticancer drug photorelease explored a high external spatiotemporal control concept for cancer treatment applications, improving localized therapy and efficiency [351].

### 8.2. Environmental Control and Remediation

Photocatalysis accelerates a reaction via catalyst–light interaction to produce electron–hole pairs for the generation of free and hydroxyl radicals, which can degrade compounds [352]. The degradation of organic pollutants in water via photocatalysis is an attractive alternative compared to conventional biological treatments, because it can be activated by innocuous light or solar radiation, presenting high efficiency [353]. Light-stimulated micromotors can accelerate the efficiency of photolysis via light-activated motion, demonstrating high potential for biodegradation applications (Figure 1(Bg)) [354]. For example, Dong et al. developed a bismuth oxyiodide (BiOI)-metal JM approach for self-electrophoresis motion in water via visible-light irradiation. BiOI microspheres are semiconductor materials that have higher photocatalytic activity for compound degradation [355]. BiOI microspheres of 2~4 µm were synthesized via the solvothermal method from bismuth nitrate oxide, potassium iodide, and ammonium hydroxide. Uniform particle monolayers were sputter-coated with a thin gold layer of 20 nm to obtain BiOI-based JMs. Visible-light irradiation produced electrons from the BiOI side, which were trapped in the metal side, generating a negative net charge. Simultaneously, the BiOI surface rusted the water, forming H^+^ ions. The H^+^ ions migrated across the micromotor double layer to attain electric balance in the metal side, but were consumed to complete the photochemical reaction. Water electroosmotic flow was concentrated in the metal side with the H^+^ ions’ net flux, producing the self-electrophoresis motion with a speed of 1.62 μm s^−1^. BiOI self-propelled micromotors proved to be a promising fuel-free motion strategy using visible light and water synergy, which could be activated under sunlight for environmental applications (Figure 10C) [346].

Moldable organic polymeric systems have demonstrated biodegradability and solubility in aqueous media—features amenable for environmental applications, promoting high interaction with pollutants and salts compared with precious metal systems with limited modularity and worse properties [347]. Moreover, semiconductor polymers can exert photocatalytic activity for pollutant degradation, constituting another approach for water purification [356]. Kochering et al. demonstrated a visible-light-activated organic sulfur semiconductor and nitrogen donor/acceptor polymeric micromotor with self-diffusiophoresis motion for the degradation of toxic organic pollutants. A Pd-catalyzed Stille cross-coupling reaction was used to synthesize a polymeric micromotor with a mean diameter of 6.5 μm, employing a benzotrithiophene-based monomer, 3s-triazine, and Pd(PPh_3_)_4_ as precursors. Visible-light irradiation at 560 nm started the motion with a speed of 1.82 ± 0.23 μm s^−1^ via photogeneration of excitons (electron–hole pairs). The visible irradiation photocatalyzed the micromotors, causing the disaggregation of the polymer because the material absorbed the light in a situated energy gap, causing the donor/acceptor dyes to break apart. This degraded the organic pollutants and increased the acidity of aqueous media, changing their color, showing great potential for colorimetric-based acidity sensors, and opening new environmental control options and novel opportunities for remediation applications (Figure 10D) [347].

### 8.3. Biosensing and Cancer Therapeutics

Biosensors based on motion can detect biomolecules in the cellular microenvironment, such as lipopolysaccharides, endotoxins, caspase-9 [357,358], etc. Biomolecule detection is essential for efficient diagnosis and subsequent treatment [359]. Gram-negative bacteria are present in cellular infections, and their real-time detection is necessary in order to enable efficient treatment [360]. On the other hand, caspase-9 regulates apoptosis in the cell death process, and its insufficient presence in cells can lead to the development of cancer [361]. Light-stimulated systems offer remarkable opportunities for the real-time optical detection of biomolecules via light exposition [362]. For example, Liu et al. developed a UCNPs system for the detection of caspase-9, integrating a dye-functionalized specific peptide. NIR irradiation activated the UCNPs for the emission of green light; the dye in the nanoconjugate absorbed the green light, but when the intracellular caspase-9 was cleaved, the dye peptide was detected via the light, sensing the presence of caspase-9 [363].

Light-stimulated micromotors enhance the biosensing detection kinetics, because their motion improves the sensor–analyte interaction (Figure 1(Bg)) [364]. For example, Jurado-Sanchez et al. reported a fluorescent phenylboronic acid (PABA)-loaded JM for the detection of lipopolysaccharides, using graphene quantum dots (GQDs); Pt nanoparticles were responsible for catalytic motion, with H_2_O_2_ as a fuel, and magnetite nanoparticles for magnetic motion controlled by an external magnetic field. The JMs were prepared via the nanoemulsion methodology, using an oil–water combination, and the magnetite and Pt nanoparticles were self-situated in the corners of the nanoparticles because of the change in their solubility during the polymeric particle formation process. Interaction between GQDs and the target endotoxin produced fluorescence emission quenching on the GQDs, due to the PABA recognizing the lipopolysaccharide receptors interfering with the emission of the GQDs. This specific sensing approach paved the way toward creating lab-on-chip technologies for biomedical and biological applications (Figure 10E) [348].

Using biopolymeric nanomotors for disease intervention is a hot topic, because they are a proof-of-concept paving the way toward an actual application [365]. Biomedical intervention needs a fuel-free or biological fuel system to generate motor motion in biological environments [20], while intracellular applications require carriers at the nanoscale for efficient cellular interaction [366]. Therefore, biologically fueled, light-stimulated polymeric nanomotors may fulfill both requirements (Figure 1(Bg)) [12,367]. Choi et al. developed peroxide-fueled NIR thermal polymeric JMs for photothermal therapy, using a photoresponsive naphthalocyanine-loaded PEG-block-PEO. A combination of a nanoprecipitation methodology and a dialysis technique formed nanovesicles with Pt NPs in the core for catalytic movement in the presence of small amounts of H_2_O_2_ from the cancer cells. NIR irradiation activated the polymer, increasing its temperature and changing its character from hydrophilic to hydrophobic, supporting controlled motion and enhanced photothermal therapy (Figure 10F) [349].

In summary, this section showed a variety of fuel-based and fuel-free micro/nanomotors with the ability to generate motion based on thermophoresis, magnetic interaction, catalytic reaction, electrophoresis, diffusiophoresis, the Marangoni effect, and photoisomerization effects, demonstrating remarkable versatility in various applications, as summarized in Table 4.

## 9. Current Challenges, Opportunities, and Concluding Remarks

This review showed the principal strategies to synthesize photosensitive polymeric backbones to form various micro/nanocarrier structures, including micelles, polymersomes, and spheres. The choice of carrier type depends upon cargo characteristics, polarity, molecular size, biocompatibility, and functional groups, which could open up countless opportunities for cargo release. The photoactive molecules were strategically incorporated into the carriers in order to respond to light irradiation, and had spatiotemporal control over the kinetic cargo release. The principal studies on PCs have been employed in biomedical applications, pointing to the need to control cargo release—e.g., drugs, DNA, or photosensitizers—in order to increase the efficiency and efficacy of diagnosis and therapeutic regimens. The cellular specificity of functional PCs helps to decrease cargo concentration, while maintaining their therapeutic effect(s). However, directing site-specific PCs has not been sufficiently explored.

The fast cargo release of these nanosystems has already been demonstrated, showing great promise as a drug delivery strategy. However, thorough testing of the increased therapeutic effect(s) of PCs in various diseases merits further exploration, indicating that this biophotonic field is in the first stage toward the development of actual treatments—especially as a principal intervention for cancer treatment. Cytotoxicity, genotoxicity, and phototoxicity studies of the subproducts delivered intra- and extracellularly after UV and NIR photostimulation must be investigated in depth; it is necessary to understand the cellular biological mechanism(s) of their degradation, and to verify their biocompatibility and biodegradability. Furthermore, new photoactive molecules must be discovered or developed in order to decrease photostimulation time, which is a considerable challenge for the improvement of biophotonic applications. PCs’ versatile structures can provide cargo loading for multipurpose applications and, when functionalized with ligands, facilitate biological interactions such as cellular affinity and furtivity. This versatility demonstrates their useful features for theragnostic applications, with considerable interest in developing the next generation of nanomedicines.

Producing the self-motion of micro/nanocarriers is a cutting-edge research topic. Such a challenge is limited only by creativity; wherein unprecedented results expect to offer external control over the kinetics involved in the cargo release process. After reviewing the light-stimulated polymeric micro/nanomotors investigated in previous years, external control of photo-thermophoresis was shown to be the most important mechanism for motion generation. However, it is necessary to explore new forms in order to produce fast photomotion alternatives, departing from gold as the principal source, creating new options, and improving the versatility of the micro/nanomotors. The principal challenge is in developing soft chemistry approaches for the synthesis of self-fueling micro/nanomotors, and improving their reproducibility and the possibility of motion in various media. Moreover, nanomotors’ intracellular motion for enhanced biomedical intervention at the nanoscale could soon impact theragnostic duties.

Furthermore, NIR irradiation is the future of the field of biophotonics in biomedical intervention, because of its biocompatibility and high tissue penetration characteristics; it opens up opportunities to create new devices for NIR generation and, thus, take better advantage of this energy.

## Figures and Tables

**Figure 1 polymers-13-03920-f001:**
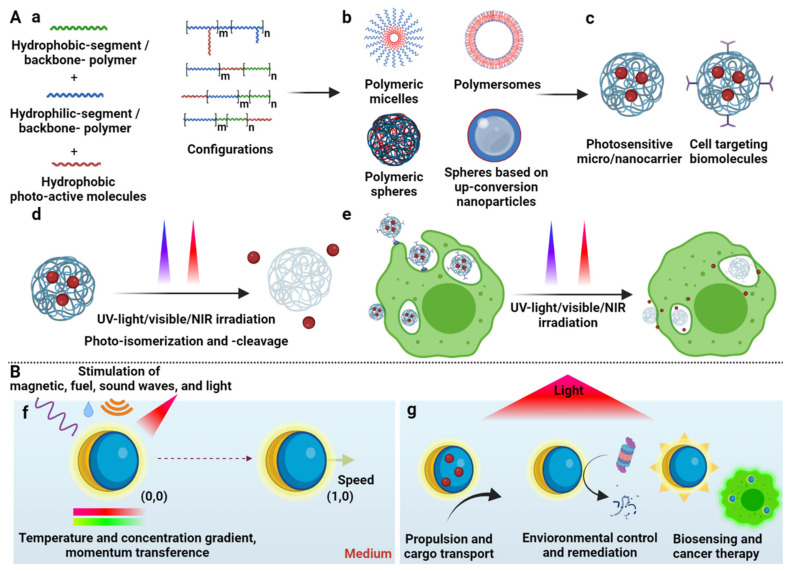
Schematic illustration of polymeric (**A**) micro/nano-carriers and (**B**) motors for cargo transport and phototriggered delivery. Photosensitive polymer synthesis (**a**), photosensitive polymeric micro/nanocarriers (**b**), and polymeric micelles are represented by red hydrophobic and green hydrophilic segments; polymersomes by blue hydrophilic and red hydrophobic segments; polymeric solid spheres by a red backbone polymer dispersion; and spheres by gray upconversion nanoparticles, blue polymer, and red photoactive molecules. Cargo transport and functionalization for targeting cells (**c**); cargo photorelease concept (**d**); cellular environment for biomedical applications (**e**); cargo concept (red) and cell-targeting biomolecules (purple), polymeric micro/nanomotors (**f**), and light-stimulated polymeric micro/nanomotor applications (**g**).

**Figure 2 polymers-13-03920-f002:**
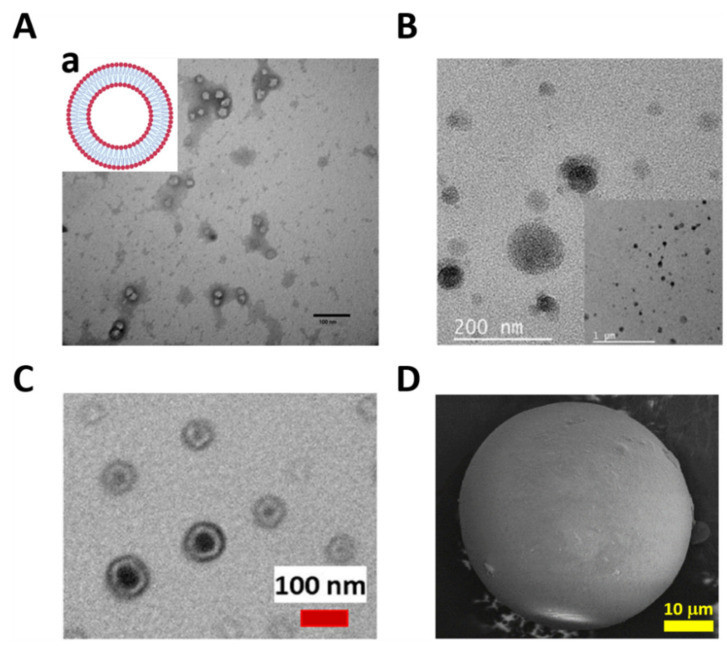
Schematic illustration of different types of micro/nanocarriers: TEM images of (**A**) liposomes based on a red hydrophobic head and blue hydrophilic segments (**a**), reprinted with permission from [99], copyright 2017, Springer Nature; (**B**) polymeric nanomicelles, reprinted with permission from [100], copyright 2021, Penske Media Corporation; (**C**) nanopolymersomes, reprinted with permission from [101], copyright 2020, Springer Nature, and SEM images of (**D**) polymeric microspheres, reprinted with permission from [102], copyright 2021, Elsevier.

**Figure 3 polymers-13-03920-f003:**
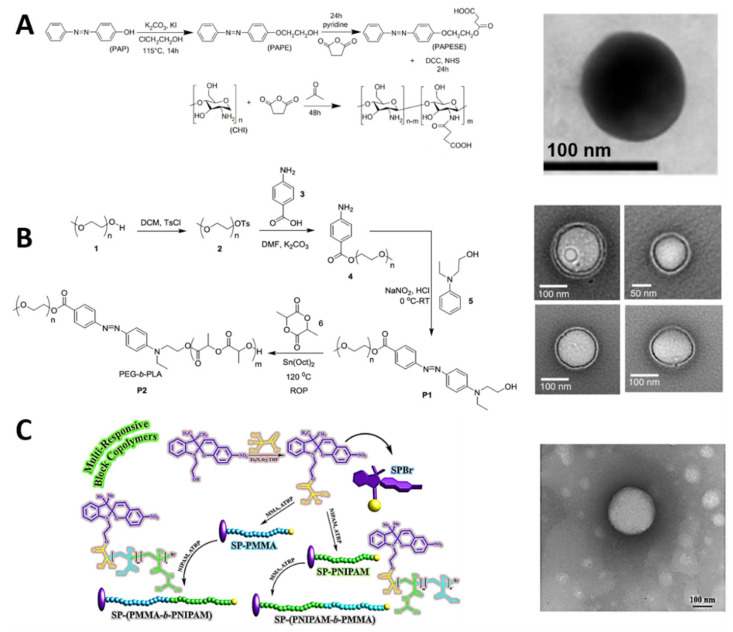
Schematic synthetic route (**left**) and TEM images (**right**) of the corresponding UV photoisomerizable polymers and nanocarriers. (**A**) PNSC-copolymer, reprinted with permission from [146], copyright 2020, Springer Nature; (**B**) azobenzene PEG-b-PLA-copolymer, reprinted with permission from [147], copyright 2018, Springer Nature; and (**C**) SP-(PMMA-b-PNIPAM)-copolymer, reprinted with permission from [148], copyright 2021, Elsevier.

**Figure 4 polymers-13-03920-f004:**
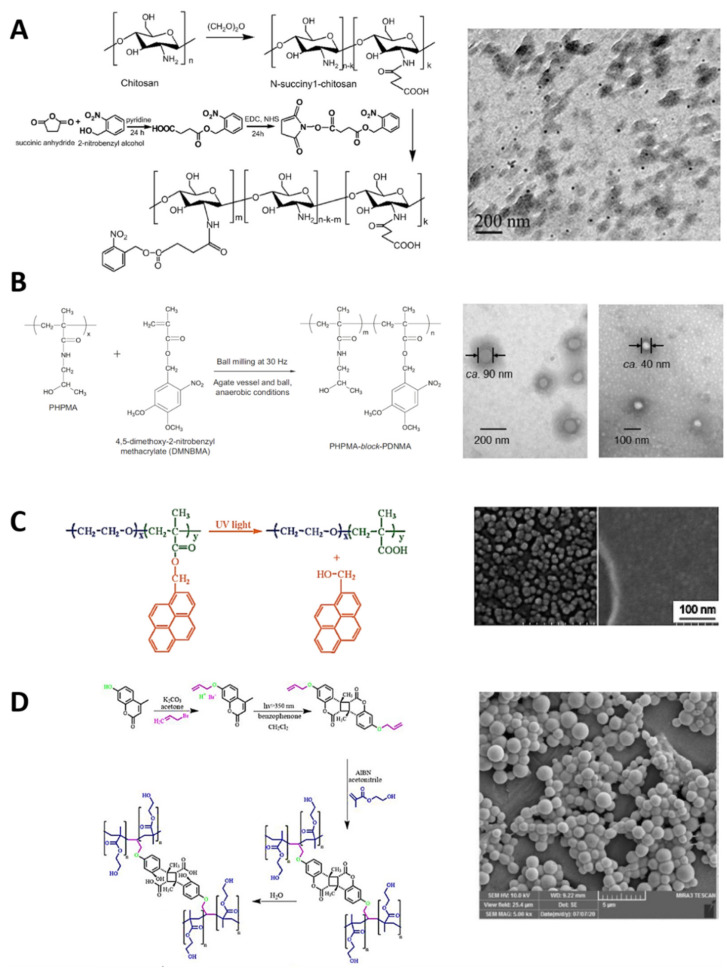
Schematic synthetic route (left) and TEM images (right) of the corresponding UV–NIR photocleavable polymers and carriers. (**A**) o-nitrobenzyl APNSC, (**B**) o-nitrobenzyl PHPMA, (**C**) pyrenylmethyl ester, and (**D**) coumarinyl ester copolymers, respectively. Reprinted with permission from [149,150,151,152], Copyright 2013, Elsevier, 2020 Springer Nature, 2005 American Chemical Society, and 2021, Elsevier.

**Figure 5 polymers-13-03920-f005:**
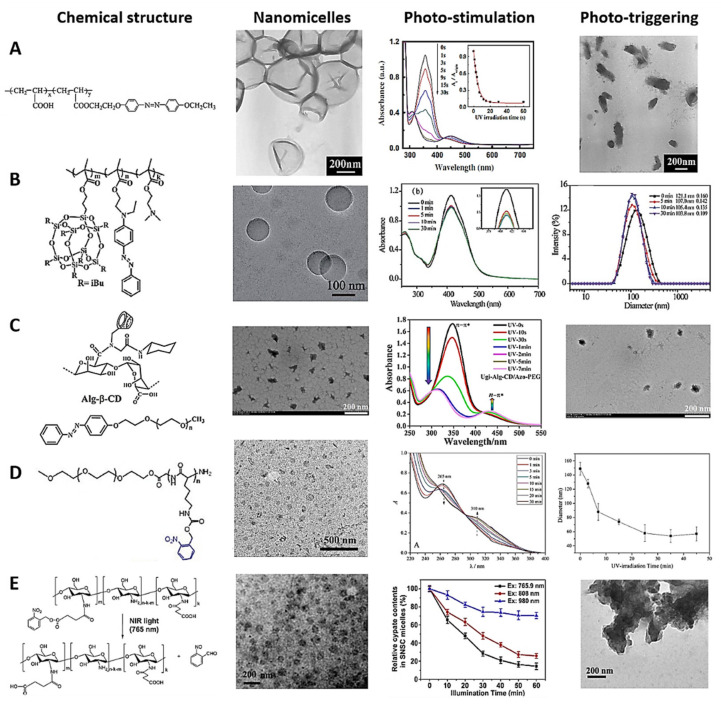
Chemical structure, TEM images, photostimulation, and phototriggering characterization of photosensitive nanomicelles. Study of PNMs based on (**A**) PPAPE, (**B**) copolymer based on methacrylate isobutyl polyhedral oligomeric silsesquioxane and azobenzene derivatives, (**C**) amphiphilic Pickering emulsion, (**D**) poly(o-nitrobenzyloxycarbonyl-L-lysine)-b-PEO, and (**E**) o-nitrobenzyl N-succinyl chitosan polymer. Reprinted with permission from [149,180,211,212,213]. Copyright 2012, Elsevier; 2018, Royal Society of Chemistry; 2021, Elsevier; 2017, Royal Society of Chemistry; and 2013, Elsevier, respectively.

**Figure 6 polymers-13-03920-f006:**
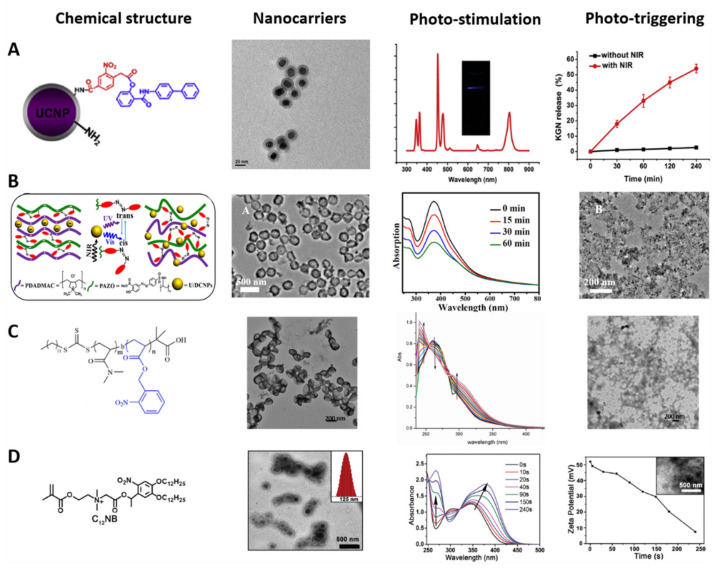
Chemical structure, TEM and AFM images, photostimulation, and phototriggering characterization of photosensitive nanocarriers: (**A**) photosensitive-nanospheres based on UCNPs, reprinted with permission from [220], copyright 2016, Elsevier; (**B**) photosensitive layer-by-layer polymer, reprinted with permission from [221], copyright 2018, John Wiley and Sons; (**C**) o-nitrobenzyl polymer, reprinted with permission from [140], copyright 2020, Multidisciplinary Digital Publishing Institute; and (**D**) terminal o-nitrobenzyl polymer, reprinted with permission from [222], copyright 2020, Royal Society of Chemistry.

**Figure 9 polymers-13-03920-f009:**
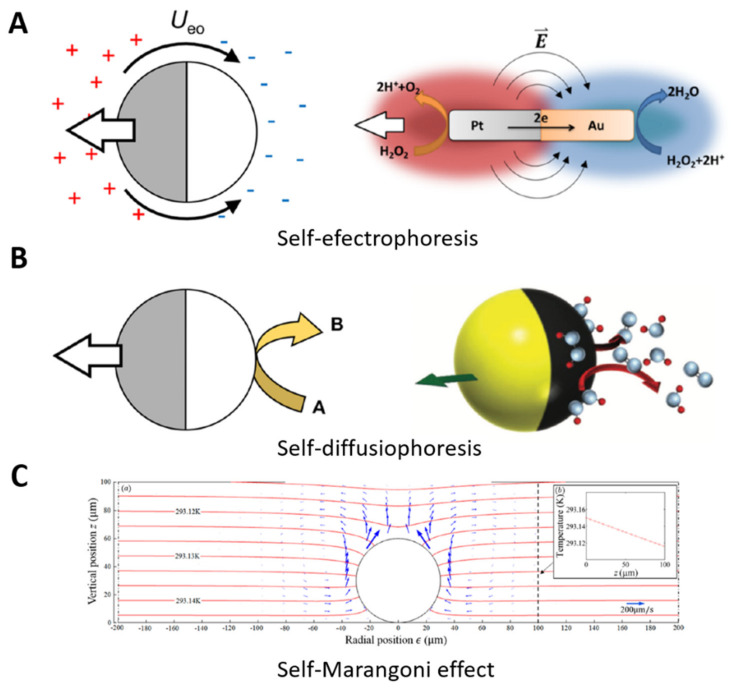
Schematic representation of micro/nanomotors’ propulsion mechanisms: (**A**) self-electrophoresis; (**B**) self-diffusiophoresis, reprinted with permission from [322], copyright 2019, American Chemical Society; and (**C**) Marangoni effect, reprinted with permission from [325], copyright 2015, American Chemical Society.

**Table 2 polymers-13-03920-t002:** Characterization of photosensitive micro/nanocarriers.

Characterization Parameter	Method	Comment	Ref.
Morphological and physicochemical	FT-IR	The absorption spectroscopy technique works via application of a polychromatic infrared beam over a compound in liquid, solid, or gaseous state.	[159]
		The infrared spectrum is obtained via the Fourier transform mathematical process, in a wavenumber range between 4000 cm^−1^ and 660 cm^−1^.	[160]
		Chemical functional groups—such as carboxylic, hydroxy, and amine groups, among many others—absorb and transmit infrared light at different wavenumbers, characterizing the PMNs’ organic composition.	[161,162]
	NMR	Structural characterization technology in which an oscillating magnetic field disturbs a nucleus molecule, producing a particular electromagnetic signal.	[163]
		The frequency of this signal depends on the chemical structure and nonzero nuclear spin of the involved isotope.	[164]
		^1^H NMR is a spectrometric method that uses the magnetic moment and angular momentum of the hydrogen-1 nucleus to obtain its frequency characteristics, which depend on the atom–proton interaction, such as C–H and O–H, determining the molecular structure of the PMNs.	[165,166]
	TEM	The micro/nanoscale image capture method detects the transmitted electrons after irradiation with an electron beam over a target sample.	[167]
		The staining technique from TEM permits differentiation between nanopolymersomes, nanomicelles, and nanospheres, because the hydrophobic core and interface of the micelles and polymersomes are stained for differentiation via dark electron absorption (Figure 2B,C, respectively).	[168,169,170]
	SEM	Electron-beam-based technique that scans the sample’s surface via electron–atom interaction, characterizing its topography and composition.	[171]
		The spheres with a high density of hydrophobic and hydrophilic segments can be visualized via SEM (Figure 4D).	
Morphological and physicochemical	TEM and SEM	Characterizes morphology, size, and elemental composition of the PMNs.	[172]
	DLS	Uses an optical mode to quantify the scattered light beam for the particles dispersed in the medium, characterizing the size distribution of PMNs, concerning the sample’s intensity extent, volume, or number.	[173,174]
	SLS	Estimates the molecular weight of NPs in the solution.	[175]
	ELS	Describes the static electric field intensity on the double-layer limit from a particle/molecule dispersed in a fluid.	[176]
		The superficial charge known as ζ-potential determines the PMNs’ dispersity and the electron density characteristics of the available groups on the nanoparticles’ surface.	[177]
	Fluorescence spectroscopy	Quantifies a compound’s light emission after electron excitation via light beam irradiation.	[178]
	Absorbance spectroscopy	Spectrophotometric method characterized by wavelength-dependent absorption of a compound after light irradiation.	[179]
		Absorbance spectroscopy describes the photoisomerization and photocleavage mechanism by absorbing the PAMs’ characteristic wavelength peaks.	[154,156]
	Fluorescence and absorbance spectroscopy	Allows tracking and quantification of the presence of fluorescence and absorbance of loaded and photoreleased cargo in the PMNs, such as Nile red, doxorubicin, Dil, etc.	[147,157,180]
Cargo LC, EE, and photorelease	Gravimetry, titrimetry, and potentiometry	Employed to quantify the concentration of a compound photoreleased from PMNs.	[181]
	Gravimetry	Quantitative measurement of the solid analyte weight to concentration determination.	[182]
	Titrimetry	The quantitative chemical analysis uses neutralization reactions to measure a solution’s concentration via volumetric or potentiometric techniques.	[183]
		Interaction between the solutions involved in the reactions and an indicator that generates a color change characterizes the volumetric techniques to calculate the volumes’ concentration.	[184]
	Potentiometry	Employs the turning points of the pH–volume curve between the target and the known solution in order to determine the hydrogen cation concentration and, indirectly, the concentrations of some compounds.	[185]
Cargo LC, EE, and photorelease	Chromatography	Methods characterized by a physical separation, employing the mobile and stationary phases of the molecules involved in the process in gas and liquid states to quantify their concentrationa.	[186,187]
		High-performance liquid chromatography (HPLC) quantifies cargo concentration by separating mobile and stationary liquid phases in order to establish the LC and EE.	[188]
	Spectrophotometry and fluorescence spectroscopy	Commonly used to quantify the cargo concentration via UV–Vis absorption and fluorescence intensity techniques, using the cargo absorbance or fluorescence curves regarding concentration and weight—either directly in the loaded cargo or indirectly in the supernatant (free cargo)—and then determining the LC and EE.	[189]
		Spectrophotometry is the most commonly used method to quantify the delivered cargo, because of its straightforward use, taking advantage of the cargo absorption and fluorescence properties, as characterized by UV–Vis absorption and fluorescence techniques.	[190,191]
Movement	Tracking analysis	Motion videos are taken by high-resolution cameras adapted to optical and fluorescence microscopy. They are then analyzed using tracking software or programs such as Image J to quantify the velocity and distance traveled by the particles.	[192,193,194]
	Nanoparticle trace analysis (NTA)	Nanoparticles’ motion is analyzed in solution using NanoSight for particles between 10 and 1000 nm in size. Each particle is analyzed individually using NTA, via direct observation and measurement of diffusion events.	[195,196]

**Table 3 polymers-13-03920-t003:** Photosensitive micro/nanocarriers for biomedical applications.

Type	PAMs	Features	Application	Ref.
Polymeric nanomicelles	Azobenzene	30s/UV light/365 nm/photoisomerization	Concept of cargo photodelivery	[211]
		5 min/UV light/365 nm/photoisomerization	Hydrophobic model drug (Nile red) photorelease	[180]
		UV light/365 nm/photoisomerization	Hydrophobic model drug (Curcumin) photorelease	[212]
		14 s/UV light/365 nm/photoisomerization	Concept of specific cargo photorelease into cells	[146]
	O-nitrobenzyl	30 min/UV light/365 nm/photocleavage	Drug/gene/protein photodelivery	[213]
Polymeric nanomicelles	O-nitrobenzyl	60 min/NIR irradiation/765nm/photocleavage	Drug model photorelease in deep tissues	[149]
Liposomes	Azobenzene	UV light/365 nm/photoisomerization	Doxorubicin delivery	[226]
	O-nitrobenzyl	NIR irradiation photocleavage	Concept of cargo photodelivery	[227]
		NIR irradiation/740 nm/pcCPP-linked	siRNA photodelivery into specific cells	[249]
Phospholipid-PEG nanoparticles	Fluorophore model	NIR irradiation ≥ 1000 nm/anti-EGFR antibody-linked	NIR fluorescence in vivo imaging of MDA-MB-468 cells	[246]
Nanopolymersomes	Azobenzene	UV light/360 nm/photoisomerization	Photorelease of hydrophilic and hydrophobic molecules	[147]
	O-nitrobenzyl	UV light/365 nm/photocleavage	Photorelease of hydrophobic doxorubicin -release as potential use in cancer therapy	[140]
		UV light/360 nm/photocleavage	Doxorubicin photodelivery into HeLa cells	[222]
Nanospheres based on UCNPs	O-nitrobenzyl	60 min/NIR irradiation/980 nm/photocleavage	Model drug delivery into cells for enhancing neocartilage formation in vivo	[220]
		NIR irradiation/980 nm/adamantane-dimer-targeting peptide-linked	Drug photodelivery into cancer cells	[247]
Polymeric nanospheres	Verteporfin photosensitizer	NIR irradiation/690 nm/hTfr peptide-linked	Phototriggered treatment of breast cancer	[248]
	^D^TRCD	NIR irradiation/660 nm/ROS photogeneration	Specific drug delivery into cell nuclei and enhanced cancer therapy	[250]

**Table 4 polymers-13-03920-t004:** Propulsion mechanisms, features and applications of micro/nanomotors.

Motor	Propulsion Mechanism	Features	Application	Ref.
Janus capsule based on silica particles and gold	Thermophoresis triggered by light	NIR irradiation/808 nm/1–20 μm diameter/42 μm s^−1^ speed	Dynamic strategy for cancer therapy	[301]
Polymeric tubular rocket		NIR irradiation/780 nm/10 μm length and 5 μm width/160 μm s^−1^ speed	Fuel-free and externally controlled propulsion for biomedical applications	[302]
Polymeric Janus microcapsules	Magneto-catalytic reaction triggered by the Pt–H_2_O_2_ system	~150 μm diameter/200–2400 μm s^−1^ speed	Potential usefulness of soft micromotors for targeted delivery of cargo	[310]
Janus polymeric sphere	Catalytic reaction from the Mg–HCl system	~20 μm diameter/~120 μm s^−1^ speed	Antibiotic drug delivery for bacterial burden reduction in the murine stomach	[311]
Tubular polymersome	Catalytic reaction from the catalase–H_2_O_2_ system	300 nm size	Hydrophobic and hydrophilic model drug delivery	[313]
Protein- and phospholipid-based Janus droplets	Hydrolysis reaction from the urease–urea and lipase system	22–80 µm size/2–41 µm s^−1^ speed	New Janus particle formation via fusion mechanism	[318]
Self-fueled lipase-active oil droplets	Buoyancy mechanism motion	10 of 60 µm of oil droplets/membrane thickness of 7.46 nm	New lipase droplet for protocell buoyancy-induced motion	[319]
Polymeric Janus stomatocyte	Catalytic reaction from the Pt–H_2_O_2_ system	345 nm diameter/~35 μm s^−1^ speed	Cargo delivery in a cellular environment	[312]
Nanorods	Electrophoresis triggered by UV or visible light	3 μm length and 150 nm radius/21 ± 4 μm s^−1^ speed	Propulsion concept	[323]
Janus Ag-Cl	Diffusiophoresis	2.5 μm diameter	3D cargo transport and sensing applications	[324]
Tubular	Marangoni effect by the surfactant at tubular surface	Enzyme delivery	Pollutant degradation	[326]
Polymeric sphere	Photoisomerization triggered by light	UV and visible irradiation/16 nm diameter/15 μm s^−1^ speed	Light-stimulated propulsion	[341]
Janus polymersome	Thermophoresis stimulated by NIR irradiation	100 nm diameter/6.2 μm s^−1^ speed	Doxorubicin delivery into HeLa cancer cells	[345]
Janus polymersome	Catalytic reaction caused by Pt–H_2_O_2_	749.2–2634.2 nm diameter/2–90 μm s^−1^ speed	Drug photodelivery	[350]
Magnetic chitosan microswimmers	Magnetic motion at 10 mT and 4.5 Hz	3.34 ± 0.71 μm s^−1^	Doxorubicin photocleavage release	[351]
Janus BiOI microspheres	Electrophoresis stimulated by visible light	2~4 μm diameter/1.62 μm s^−1^ speed	Fuel-free motion strategy	[346]
Organic polymeric systems	Photocatalysis of semiconductor polymers under visible light	6.5 μm diameter/1.82 μm s^−1^ speed	Colorimetric-based acidity sensors	[347]
Polymeric Janus microparticle	Magneto-catalytic motion from the Pt–H_2_O_2_ system	Fluorescent phenylboronic acid and graphene-modified GQDs	Detection of lipopolysaccharides for biomedical and biological applications	[248]
Polymeric Janus nanoparticle	Catalytic reaction from the Pt–H_2_O_2_ system	NIR thermal polymer	Photothermal therapy	[349]

## Data Availability

Not applicable.

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
