# Peer review of "Polymeric Micro/Nanocarriers and Motors for Cargo Transport and Phototriggered Delivery"

_polymers, 2021, doi:10.3390/polym13223920_

Round 1

Reviewer 1 Report

This review is focused on the polymeric micro/nanocarriers and motors for cargo transport and photo-triggered delivery applications. It is an interesting topic. The manuscript is very well organized and written. I contain enough literature. I have just one minor comment on the resolution of Fig. 3C and 4D is low.

Author Response

Dear reviewer,

Please find the response to reviewers on page 54 after the revised edition of the review.

Best regards,

Reviewer 2 Report

Review report of “Polymeric Micro/Nanocarriers and Motors for Cargo Transport and Phototriggered Delivery” by Mena-Giraldo et al.

The authors reviewed recent progress in the polymeric micro/nanocarrier development based on nanomicelles, nanospheres, and nanopolymersomes, with enhanced properties to increase cargo protection and cargo release efficiency triggered by UV and NIR irradiation, including light-stimulated polymeric micromotors for propulsion, cargo transport, biosensing, and photo-thermal therapy. The way of presentation is good. I consider that the study carried out by the authors is of the interest for the readers of “Polymers” Journal, but to be accepted after minor revision. I hope the authors will revise the manuscript with utmost interest according to the following comments.

  1. The authors should consider the potential papers on the smart polymers and their interactions with co-solvents (Journal of Colloid and Interface Science 485 (2017) 183–191; Phys. Chem. Chem. Phys., 2018, 20, 9717-9744; Journal of Colloid and Interface Science 541 (2019) 1–11)
  2. Please revise the lines 45-47.
  3. Provide the articulation and distillation of cited articles of section 3.2.

Author Response

(The authors gave the same response as above.)

Reviewer 3 Report

In this manuscript, Giraldo and Orozco summarized the progress of drug delivery systems based on static and dynamic polymeric materials, ranging from different sizes and different structures, including micelles, vesicles, etc. And the emphasis was finally put on the photo-sensitive polymers and their applications. From this work, it is easy to see Orozco and coauthor have deep and comprehensive understanding about this research field, showing clear insights in this work. However, some issues are still needs to be addressed before the paper was published.

1, For the topic of this paper, the reviewer worries it is too broad, can the authors focus on the photosensitive SD-PCs? Or Polymeric micro/nanomotors for bioapps? 

2, ‘The first cargo release system and propulsion mechanism explored were cargo diffusion across the carriers and fuel-micro/nanomotors interaction, requiring longer delivery times but producing a synergic effect among chemical reactions for the dynamic motion’, in this paper, the first reference should be cited, not two review papers of ref.30-31.

3, The structures of Pyrenylmethyl ester in Table 1, seems incorrect, please revise. Similarly, the products of Coumarinyl ester seems wrong. 

4, SOme references discussing photo-sensitive motors for drug delivery are missing, like: Molecular Membrane Biology, 2010; 27(7): 364–381; Small, 2015, 11, 3762–3767; ACS Nano 2018129617, etc. Similarly, some large vesicles or protein-based capsules could be discussed as well, such as Marangoni effect induced motion (Angew. Chem. Int. Ed. 2020, 59, 16953–16960.); buoyance induced motion (Angew. Chem. Int. Ed., 2019, 58, 1067.), which could fulfill the topic of this review.

5, Some enzyme powered papers could be talked in the background, such as urease, catalase and lipase, etc. due to their nice biocompatibility.

Author Response

(The authors gave the same response as above.)

Round 2

Reviewer 3 Report

Many thanks for the authors' revision. Almost all of the issues proposed by the reviewers were well addressed. Thereby, the paper could be considered for publication.

Only one little worry left:

For the formula in table 1E, I appreciated the authors provide the original references of ‘Biomacromolecules 2018, 19, 1840–1857; and Polymers 2021, 13, 2464’. Although these formulas are presented in the referenced papers (also reviews), it is still strange for 1E formula. And I further checked the original papers (Ref.41-43) cited by ‘Biomacromolecules 2018, 19, 1840–1857; and Polymers 2021, 13, 2464’, which can not confirm the 1E reaction formula is correct. So, before the final publication, could the authors double check more references to confirm what should be the products of 1E reaction? Because the ester decomposition product is hard to obtain one more carbon atom to the phenol rings (like the first product of 1E equation). Please double check. 

Author Response

Dear reviewer,

Please find the response to reviewers in the attachment after the revised edition of the review.

Best regards,
